



# Simulating the Holocene evolution of Ryder Glacier, North Greenland

Jamie Barnett[1,2], Felicity A. Holmes[1,2], Joshua Cuzzone[3], Henning Åkesson[4], Mathieu Morlighem[5], Matt O'Regan[1,2], Johan Nilsson[2,6], Nina Kirchner[2,7,8], and Martin Jakobsson[1,2]

[1]Department of Geological Sciences, Stockholm University, Sweden
[2]Bolin Centre for Climate Research, Stockholm University, Sweden
[3]Joint Institute for Regional Earth System Science and Engineering, University of California, Los Angeles
[4]Department of Geosciences, University of Oslo, Norway
[5]Department of Earth Sciences, Dartmouth College, USA
[6]Department of Meteorology, Stockholm University, Sweden
[7]Department of Physical Geography, Stockholm University, Sweden
[8]Tarfala Research Station, Stockholm University, Sweden

**Correspondence:** Jamie Barnett (jamie.barnett@geo.su.se)

**Abstract.** The Greenland Ice Sheet's negative mass balance is driven by a sensitivity to both a warming atmosphere and ocean. The fidelity of ice-sheet models in accounting for ice-ocean interaction is inherently uncertain and often constrained against recent fluctuations in the ice-sheet margin from the previous decades. The geological record can be utilised to contextualise ice-sheet mass loss and understand the drivers of changes at the marine margin across climatic shifts and previous extended warm

periods, aiding our understanding of future ice-sheet behaviour. Here, we use the Ice-sheet and Sea-level System Model (ISSM) to explore the Holocene evolution of Ryder Glacier draining into Sherard Osborn Fjord, Northern Greenland. Our modelling results are constrained with terrestrial reconstructions of the paleo-ice sheet margin and an extensive marine sediment record from Sherard Osborn Fjord that details ice dynamics over the past 12.5 ka years. By employing a consistent mesh resolution of <1 km at the ice-ocean boundary, we assess the importance of atmospheric and oceanic changes to Ryder Glacier's Holocene

behaviour. Our simulations show that the initial retreat of the ice margin after the Younger Dryas cold period was driven by a warming climate and the resulting fluctuations in Surface Mass Balance. Changing atmospheric conditions remain the first order control in the timing of ice retreat during the Holocene. We find ice-ocean interactions become increasingly fundamental to Ryder's retreat in the mid-Holocene; with higher than contemporary melt rates required to force grounding line retreat and capture the collapse of the ice tongue during the Holocene Thermal Maximum. Regrowth of the tongue during the neo-glacial

cooling of the late Holocene is necessary to advance both the terrestrial and marine margins of the glacier. Our results stress the importance of accurately resolving the ice-ocean interface in modelling efforts over centennial and millennial time scales, in particular the role of floating ice tongues and submarine melt, and provide vital analogous for the future evolution of Ryder in a warming climate.





# 1 Introduction

The rate of mass loss from the Greenland Ice Sheet (GrIS) has increased fivefold between 1992 and 2020, culminating in a contribution to global mean sea-level rise of $13.59 \pm 1.27$ mm, roughly double that of the Antarctic Ice Sheet over the same period (Otosaka et al., 2023). Mass loss from Greenland can be partitioned between changes in the ice sheet's Surface Mass Balance (SMB) (Box, 2013; Fettweis et al., 2020), and fluctuations at the marine interface of tidewater glaciers that drain the ice sheet's interior, known as discharge (Enderlin et al., 2014; Mouginot et al., 2019). Rising atmospheric temperatures

have been the key driver of increased ablation and runoff from the ice sheet (Box et al., 2022), while the present influence of warm Atlantic Waters (AW) has been another major driver in the retreat and acceleration of Greenland's marine outlet glaciers (Slater and Straneo, 2022). Yet there is large spatial and temporal heterogeneity in the amplitude at which both processes drive mass loss across the ice sheet (Mouginot et al., 2019). As a consequence, projections of future sea-level rise contribution from Greenland vary by an order of magnitude for high emissions scenarios (Goelzer et al., 2020). The largest uncertainties in such

predictions stem from the limited ability of ice-sheet models to accurately resolve fluctuations in ice discharge from tidewater glaciers, where calving processes dominate mass loss and are estimated to account for up to 70% of ice sheet's sea-level contribution by the end of the century (Goelzer et al., 2020; Choi et al., 2021). The fidelity of ice-sheet models in capturing such ice-ocean processes is often tested, and refined, against the satellite observation record. This implies that tidewater glacier behaviour can only be constrained across the most recent decades, when climatic conditions were distinctly cooler than what

we expect in the future. Moreover, the ocean exhibits considerable natural variability on yearly to multi-decadal time scales. This complicates and potentially skews our interpretation of how oceanic changes affects the ice sheet (Khazendar et al., 2019). The geological record can be used to extend this history over centennial and millennial time scales, to time periods similar to that of future projections. This provides valuable context for the observed ice-sheet behaviour over the last few decades, and important natural baselines and constraints for models of current and future ice-sheet change.

The Holocene interglacial period presents the opportunity to study Greenlandic glacier behaviour across three distinct phases: first, the recession of the GrIS during the deglaciation following the Last Glacial Maximum (LGM) in response to rapidly warming temperatures (11.7-8.2 ka BP); second, the response of the ice sheet to the Holocene Thermal Maximum (HTM, 8.2-4.2 ka BP), a protracted period when Arctic temperatures were $3\pm1°C$ above pre-industrial averages (Briner et al., 2016; Kaufman and Broadman, 2023); and third, the neoglacial cooling of the Holocene that may have begun already at about 6.5-6.3

ka BP (Davis et al., 2009) and leading into the Little Ice Age where the GrIS margin re-advanced from its Holocene minimum towards a pre-industrial state (Kjær et al., 2022). Terrestrial studies have presented a detailed geological record of the GrIS margin evolution across different regions (e.g. Kelly and Bennike, 1992; England, 1999; Funder et al., 2011; Larsen et al., 2019; Young et al., 2021), culminating in a Greenland-wide reconstruction of ice-margin retreat during the early Holocene by Leger et al. (2023) (PaleoGrIS). Furthermore, marine geophysical mapping and marine sediment cores have provided additional

knowledge of the timing of outlet glacier retreat and changing ice dynamics throughout the Holocene (Wangner et al., 2018; Reilly et al., 2019; Jakobsson et al., 2018; O'Regan et al., 2021). This is especially timely as paleo ice-sheet modelling efforts are using increasingly fine resolution (<3 km) in regional studies in order to accurately resolve fjord-scale glacier dynamics





**Figure 1.** A map of Northern Greenland highlighting the contemporary drainage basins of Petermann, Steensby, Ryder and Ostenfeld Glaciers (Mouginot and Rignot, 2019). Paleo ice sheet margins of the Kap Fulford (>12.5 - 10 ka BP, pink) and Kap Warming Stade (>9.5 - 8 ka BP, purple) are shown as described in the terrestrial study by Kelly and Bennike (1992). Small dashed lines indicate hypothesised marine margins of GrIS, while large dashes are used to join data gaps.

and to assess the influence of ice-ocean interactions on the GrIS throughout the Holocene (Briner et al., 2020; Kajanto et al., 2020; Cuzzone et al., 2022).

Recognising the patterns and drivers of retreat during the Holocene, the current interglacial period, is especially pertinent to the Northern Sector of the GrIS. At the LGM, the ice sheet coalesced with the Innuitian Ice Sheet (England et al., 2006) and likely extended out into the Lincoln Sea where ice streams emanating from the Nares Strait and the northern fjords likely formed ice-shelf conditions (Dawes, 1986; England, 1999; Larsen et al., 2010). During the Holocene, this marine-based ice system collapsed, as the GrIS decoupled from the Innuitian Ice Sheet and the northern outlet glaciers retreated back into their

modern day fjords. Owing to the remote nature of the region and harsh sea-ice conditions, field surveys have been sparse and sporadic (Koch, 1928; Dawes, 1977; Weidick, 1978). Based on available data in the early 1990s, North Greenland's Holocene history was summarised into a series of "Stades" (Danish for stages) by Kelly and Bennike (1992) (Fig. 1). The Kap Fulford





Stade relates to the most recent ice-sheet maximum, the Late Weichselian Glaciation, when the northern glaciers extended to the outer fjords and the GrIS and Innuitiain Ice Sheets remained coalesced in Nares Strait. The retreat from this last maximum ice-sheet extent is proposed to have occurred >10.5 cal ka BP by O'Regan et al. (2021), after combining new marine radiocarbon dates and existing terrestrial dates (Kelly and Bennike, 1992). The Warming Land Stade follows between >9.5 to 8 cal ka BP, corresponding to a well developed moraine sequence that represents a standstill of glacial retreat 20 - 60 km inland of the Kap Fulford Stade. Finally, a Steensby Stade characterises a neo-glacial re-advance of the GrIS. While the onset of this re-advance is poorly dated, it may have occurred prior to the late Holocene, perhaps as early as around 5.8 ka BP (O'Regan et al., 2021).

At present, Northern Greenland contains enough ice to raise global mean sea levels by ∼93 cm (Mouginot et al., 2019). However, mass loss had been relatively subdued from the region until the 2000s, likely due to the buttressing of inland ice provided by floating ice tongues limiting ice discharge (Hill et al., 2017; Millan et al., 2018, 2023). Since 1978, the region's ice tongues have lost ∼25% of their volume, with the notable collapse of Ostenfeld Glacier's ice tongue in 2003. Petermann, Steensby and Ryder Glaciers remain buttressed by their floating ice tongues, yet they have all seen grounding-line retreat and a distinct acceleration of ice discharge that has been explicitly linked to increasing basal melt rates beneath the floating ice (Hill et al., 2018a; Millan et al., 2023). During the Holocene, it is believed that the ice tongues of Petermann and Ryder Glacier's disintegrated and they retreated inland of the current grounding-line positions (Jakobsson et al., 2018; Reilly et al., 2019; O'Regan et al., 2021). Such behaviour could serve as a potential analogy for the future evolution of the contemporary outlet glaciers, where the collapse of their floating ice tongues may accelerate ice discharge and the region's contribution to global sea-level rise (Hill et al., 2018b; Humbert et al., 2023).

Here we use the Ice-sheet and Sea-level System Model (ISSM; Larour et al., 2012) to explore the dynamics of Ryder Glacier during the Holocene, constrained by unique marine sediment archives, mapped submarine glacier landforms and terrestrial evidence. Focusing on a single outlet glacier allows us to employ a model that resolves ice-ocean and grounding line dynamics in high resolution (<1 km). This puts more confidence in our simulations, and gives us the opportunity to provide a new, more detailed understanding of how Ryder Glacier and its ice tongue responded to a warming atmosphere and ocean.

## 2 Ryder Glacier

Ryder Glacier currently drains the GrIS into Sherard Obsorn Fjord (Fig 2), where a floating tongue extends ∼25 - 30 km from the grounding line and velocities peak at 500 m yr$^{-1}$ (Wilson et al., 2017; Hill et al., 2017). Sherard Osborn fjord continues ∼70 km from the present day grounding line towards the Lincoln Sea, reaching a maximum depth of 890 m and containing two shallow bathymetric sills; an outer (375 - 475 m below sea level) and inner sill (193-390 m below sea level), with the latter thought to be playing a key role in shielding the glacier from sub-surface warm Atlantic Waters (AW) (Jakobsson et al., 2020; Nilsson et al., 2023). The floating terminus has remained relatively stable since the 1990s, undergoing a cyclic pattern of slow advance followed by retreat, potentially controlled by fjord geometry (Holmes et al., 2021). This stability has persisted despite a retreat of the grounding line by ∼8 km since 1992 (Millan et al., 2023).



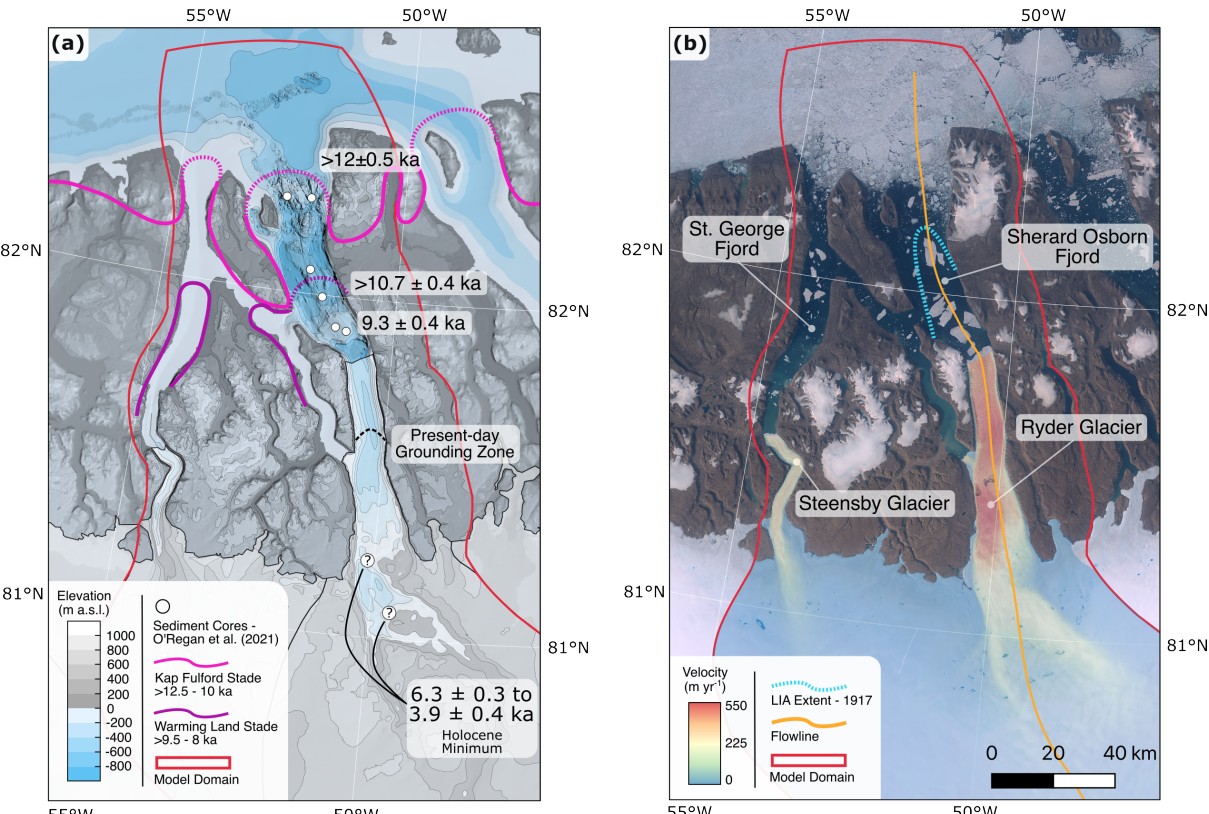

**Figure 2. (a)** A topographic map of Ryder and Steensby Glacier. Both the bathymetry and topography are taken from BedMachine v5 (Morlighem et al., 2022). Paleo ice-sheet margins of the Kap Fulford Stade (>12.5 - 10 ka BP, pink) and Warming Stade (>9.5 - 8 ka BP, purple) are redrawn from Kelly and Bennike (1992). Sediment cores taken during the Ryder Expedition on Icebreaker *Oden* are shown by white circles and are accompanied by minimum ages (O'Regan et al., 2021). **(b)** A satellite image of Ryder and Steensby glacier draining into Sherard Osborn and St. George Fjords from summer 2019 (MacGregor et al., 2020). Ice velocities are taken from MEaSUREs (Joughin et al., 2016; Joughin et al., 2018). A centre flowline for Ryder Glacier is shown in orange, the Little Ice Age extent of the glacier is highlighted in dashed blue (Koch, 1928) and the model domain used in the study is shown in red.

During the Ryder 2019 expedition with the Swedish Ice Breaker *Oden*, numerous sediment cores were taken in Sherard Osborn and have been used to describe the Holocene behaviour of Ryder Glacier by O'Regan et al. (2021) (Fig 2a). Radiocarbon dating and analysis of six lithologic units (LU) detail an evolution of the glacier that generally aligns with the terrestrial record of retreat described in Kelly and Bennike (1992). The first two lithologic units (LU6 and LU5) comprise a transition from glaciomarine diamicton to a laminated meltwater-dominated facies as the grounding line of Ryder Glacier began to retreat from the fjords' outer sill. Ages of $12 \pm 0.5$ ka BP (LU6) on the outer sill and $10.7 \pm 0.4$ ka BP (LU5) in the mid-fjord region align with the Kap Fulford Stade, while the beginning of sedimentation on the inner sill at $9.3 \pm 0.4$ ka BP (LU5) corresponds to the Warming Land Stade. A further laminated sub-ice shelf facies follows (LU4: $7.6 \pm 0.4$ to $6.3 \pm 0.3$ ka BP) before a



second diamict facies spanning $6.3 \pm 0.3$ to $3.9 \pm 0.4$ ka BP (LU3) occurs and hints at the collapse of the ice tongue during the HTM. The low sedimentation rates during this period are suggested to mark the retreat of Ryder Glacier into a small isolated
embayment, $\sim$40 km upstream of the present-day grounding line (Fig 2a). Such retreat would have reduced the delivery of fine-grained meltwater-entrained sediments but still allowed for icebergs to deposit IRD in the fjord. The following lithologic facies after $3.9 \pm 0.4$ ka BP (LU2 and LU1), inline with the Steensby Stade described by Kelly and Bennike (1992), show a transition from open-water bioturbated sediments to laminated facies that indicate the regrowth and advance of Ryder's ice tongue during the late Holocene toward the glacier's maximum in the Little Ice Age (LIA; Fig. 2b).

## 3   Methods

The finite-element thermomechanical ice-sheet model ISSM is employed to simulate the Holocene history of Ryder Glacier (Larour et al., 2012). We use the Higher-Order approximation (Blatter, 1995; Pattyn, 2003) of the Full Stokes equations for all our simulations, with our model set-up being similar to Briner et al. (2020) and Cuzzone et al. (2022), as described below.

### 3.1   Model domain and set-up

Our model domain is based on the present-day drainage catchment of Ryder Glacier, defined from velocity and slope angles of the ice surface (Mouginot and Rignot, 2019). The domain is then extended northward to include Sherard Osborn Fjord and $\sim$30 km north into the Lincoln Sea to capture LGM extend of Ryder Glacier (Fig. 2b). On the eastern side of the domain we follow the topographic divide in Wulff Land, while also extending the domain westward to include the neighbouring Steensby Glacier and Saint Georges Fjord. We do this in order to capture any affects arising from the two glaciers merging towards the Lincoln
Sea owing to the connected fjord systems. We define the base of our domain using bedrock topography from BedMachine v5 (Morlighem et al., 2022), which contains the high resolution bathymetric data of Sherard Osborn Fjord surveyed by Swedish Ice Breaker *Oden* in 2019 (Fig. 2a; Jakobsson et al., 2020).

A non-uniform mesh of the domain is produced from modern ice velocities and bedrock topography (Joughin et al., 2016; Morlighem et al., 2022). Elements range from 10 km in size for the slow-moving ice in the upper regions of the domain,
decreasing to 750 m where velocities exceed 400 my$^{-1}$ as well as in contemporary ice-free fjord areas. This ensures that grounding-line migration is simulated at resolutions of $< 1$ km for the entirety of our simulations, a resolution necessary for capturing such dynamics (Seroussi and Morlighem, 2018). The domain is then extruded vertically to contain five layers, following Cuzzone et al. (2018) we use cubic finite elements along the z-axis (prisms P1$\times$P3) for vertical interpolation to capture sharp thermal gradients near the base of the ice sheet without additional, computationally expensive, layers.

We use an enthalpy formulation from Aschwanden et al. (2012) to simulate ice temperature throughout our simulations, applying a constant geothermal heat flux at the base (Shapiro and Ritzwoller, 2004), and transient air temperatures at the surface (Section 3.4). The ice rheology parameter $B$ is initialised by solving for a present-day thermal state and allowed to vary transiently following the rate factors in Cuffey and Paterson (2010).

Basal friction is modelled through the linear viscous Budd Law (Budd et al., 1979), defining basal drag ($\tau_b$) as:



$$\tau_b = -k^2 N v_b \tag{1}$$

where $k$ is a friction coefficient, $N$ is effective pressure and $v_b$ is the basal velocity. The friction coefficient, $k$, for present day ice areas is inverted for using an adjoint method (Morlighem et al., 2010, 2013) that fits with modern velocities (Joughin et al., 2016; Joughin et al., 2018). For contemporary non-ice areas we define a friction coefficient as a function of elevation similar to Åkesson et al. (2018a):

$$k = 100 \times \frac{\min\left(\max(0, z_b + |\min(z_b)|, \max(z_b))\right)}{\max(z_b)} \tag{2}$$

where $z_b$ is the bedrock elevation relative to sea level (Morlighem et al., 2022). This produces a low friction coefficient in the deep fjords, where we would expect fast-flowing ice, and larger values across topographic highs, where ice is expected to flow slower. This spatial distribution of friction coefficients are kept fixed over time. Meanwhile, we follow Cuzzone et al. (2019) to allow the friction field to be scaled transiently based on changes in the simulated evolution of basal temperatures relative to

basal temperatures from a present-day thermal steady state of the glacier. This implementation produces an increase in friction for colder temperatures during the YD, while also rendering more slippery conditions during warmer periods such as the HTM.

We apply Dirichlet boundary conditions of ice thickness, velocity and temperature on the eastern and western boundaries of our domain. These boundary conditions are taken from a continental scale model of GrIS simulated across the same timescales with a similar model setup, albeit at a coarser resolution (Briner et al., 2020). We use different boundary conditions for each

of our SMB scenarios (Section 3.4), aligning with the experiments found in Briner et al. (2020). We justify the use of these boundary conditions as running a continental scale simulation at an equivalent resolution would be computationally infeasible. Furthermore, by using consistent boundary conditions across our model runs, we isolate the effects of our applied climate forcings on glacier behaviour. Finally, ice is allowed to freely leave the domain at the northern boundary of the domain.

All model simulations are run for 12,550 years from 12,500 BP until 2000 CE, using an adaptive time step that varies

between 0.02 and 0.3 years while satisfying the Courant-Friedrichs-Lewy criterion (Courant et al., 1928).

### 3.2 Calving parameterisation and ice-front migration

Ice-front migration is handled within ISSM using the level-set method (Bondzio et al., 2016). The ice front boundary moves at velocity:

$$\mathbf{v}_f = \mathbf{v} - (c + M)\mathbf{n}, \tag{3}$$

where $\mathbf{v}$ is the velocity vector at the ice front, $c$ and $M$ are the calving and melting rate, respectively, and $\mathbf{n}$ is the normal vector pointing outward, orthogonal to the level-set defined as a signed-distance field. During our simulations we assume that



horizontal melt at the front of an ice tongue is negligible compared to calving and ocean-induced melt from below. Thereby, we focus on applying melt rates to beneath ice that has reached flotation (Section 3.3).

A von Mises law is used to parameterise calving (Morlighem et al., 2016), where the calving rate, $c$, depends on the von

Mises tensile stress, $\widetilde{\sigma}$ and a predefined maximum stress threshold value, $\sigma_{\mathrm{max}}$ (Table 1), :

$$c = |\mathbf{v}| \frac{\widetilde{\sigma}}{\sigma_{\mathrm{max}}}. \tag{4}$$

As such, calving can only occur when $\widetilde{\sigma} > \sigma_{\mathrm{max}}$; when $\widetilde{\sigma} < \sigma_{\mathrm{max}}$ then $c = 0$. This calving parameterisation has been successfully used to simulate the migration of grounded and floating Greenlandic ice fronts in both modern (Choi et al., 2018; Morlighem et al., 2016; Åkesson et al., 2022), and paleo settings (Kajanto et al., 2020; Cuzzone et al., 2022).

### 3.3  Ocean melt

An ocean melt rate is applied to floating ice as a linear function of depth. Melt rate increases from $0$ m yr$^{-1}$ at depths shallower than 100 m to a predefined *deepmelt* value at depths below 400 m (Table 1). This implementation matches the distribution of submarine melt found through both observations and ocean modelling of Ryder Glacier (Wilson et al., 2017; Wiskandt et al., 2023), capturing the highest melt rates towards the grounding line, driven by the interplay with subglacial discharges

and warmer ocean waters, before decreasing in the cooler polar surface waters found in the upper hundred meters of the fjord. Horizontal melt at the front of the ice tongue is assumed negligible, in line with melt at depths shallower than 100 m and under the premise that frontal ablation is dominated by calving processes while submarine melt primarily acts near the grounding line beneath ice tongues (Wilson et al., 2017; Holmes et al., 2021). In the event of an ice-tongue collapse, melt is applied uniformly to the submerged grounded glacier front at half the defined *deepmelt* value in order to account for a decrease in melt towards

the water line.

### 3.4  Holocene climate

Holocene SMB is computed using a Positive Degree Day (PDD) model (Tarasov and Peltier, 1999), forced with paleoclimate reanalysis data from Badgeley et al. (2020). The model assumes that snow melts at 3 mm °C$^{-1}$d$^{-1}$ followed by ice melt at 8 mm °C$^{-1}$d$^{-1}$ for the remaining positive degree days (Braithwaite, 1995; Born and Nisancioglu, 2012; Plach et al., 2018). The

formation of superimposed ice is allowed following Janssens and Huybrechts (2000). Finally, a lapse rate of 6°C km$^{-1}$ is used to adjust the climate forcings to transient changes in the glacier's elevation.

Temperature and precipitation for the PDD model is taken from a paleoclimate reanalysis dataset produced by Badgeley et al. (2020) through the assimilation of climate model simulations and proxy data. The TraCE21Ka climate model output (Liu et al., 2009; He et al., 2013) is used in conjunction with oxygen-isotope records from eight ice cores and ice-layer thicknesses from

five ice cores to reconstruct temperature and precipitation, respectively. The resulting dataset contains a number of different temperature and precipitation pathways, varying spatially at a 50-year temporal resolution, covering the Holocene.



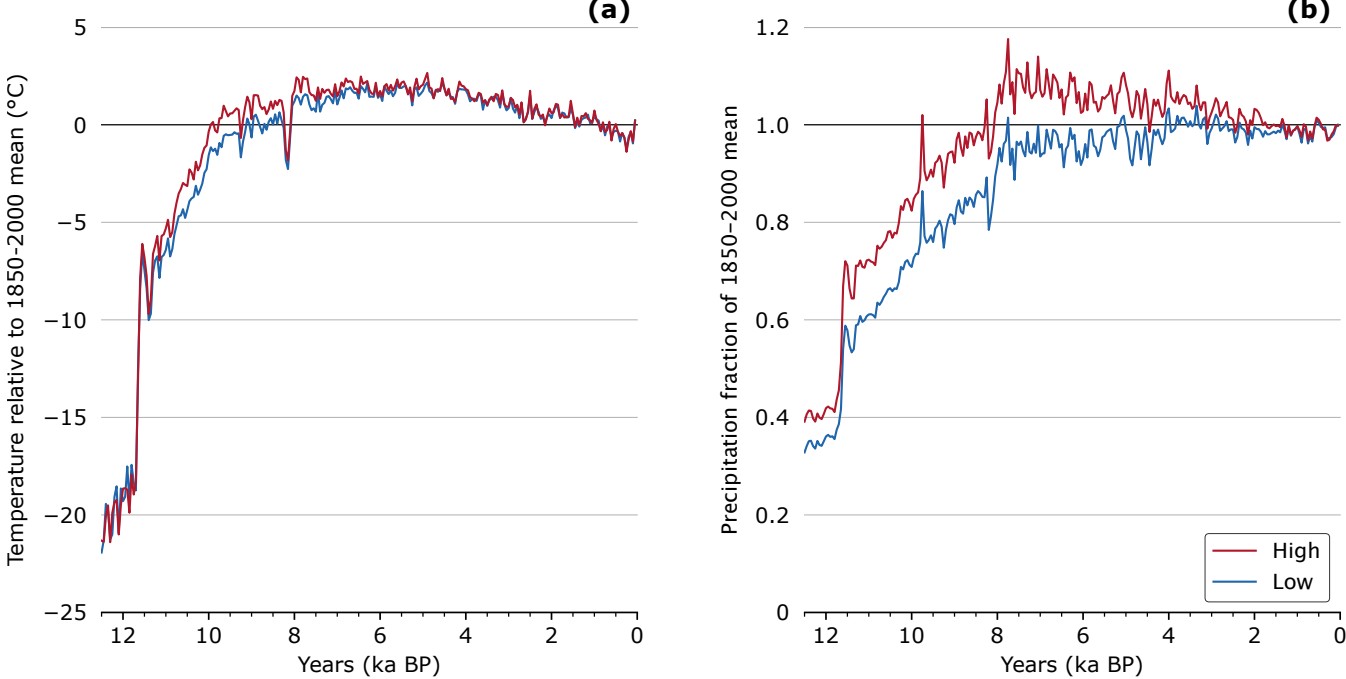

**Figure 3.** Mean annual climate data during the Holocene averaged over the model domain taken from Badgeley et al. (2020). **(a)** High (red) and low (blue) Holocene temperature scenarios expressed as anomalies with respect to a 1850 to 2000 mean (Box, 2013). **(b)** High (red) and low (blue) Holocene precipitation scenarios expressed as fractions of a 1850 to 2000 mean (Box, 2013).

We follow Cuzzone et al. (2022) in using four different Holocene SMB forcings produced from combinations of the high and low reconstructions of temperature and precipitation (Fig. 3a and b). These reconstructed variables represent the upper and lower bands of scenarios discussed in Badgeley et al. (2020) and provide a cost effective way of testing the impact of 195 both climate drivers during the Holocene. We hereby refer to each SMB scenario using the following notation: Temperature/Precipitation; where, for example, High/Low would describe the SMB scenario produced from the high-temperature and low-precipitation pathways.

The aforementioned paleoclimate variables temperature and precipitation are expressed as anomalies from a 1850-2000 mean and a fraction of a 1850-2000 mean respectively. We therefore use the comparable monthly climate variables from Box 200 (2013) to produce the final temperature and precipitation variables need for the PDD model:

$$T_t = \overline{T}_{1850-2000} + \Delta T_t, \tag{5}$$

$$P_t = \overline{P}_{1850-2000} \times \Delta P_t, \tag{6}$$



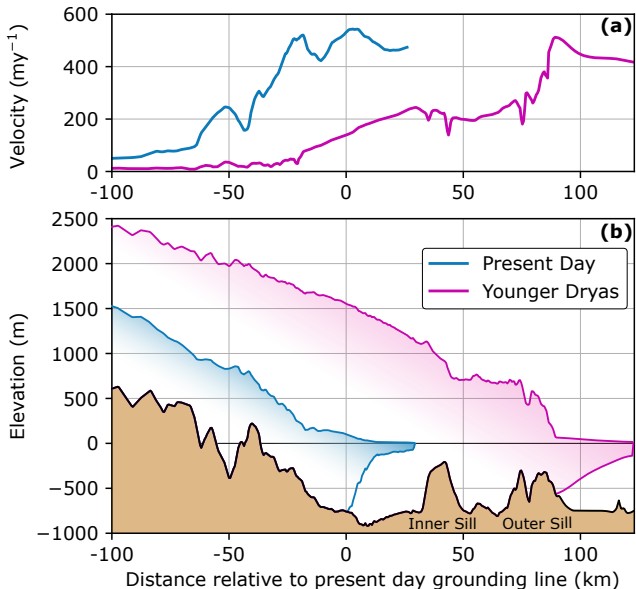

**Figure 4.** A comparison between the present day Ryder Glacier (blue) and that of the glacier after running a 20,000 year spin-up forced with a YD climate from SMB scenario Low/High (pink). **(a)** The velocity profile along the glacier's centre flowline. **(b)** A 2D cross section profile along the centre flowline. Bedrock data and the present day ice surface is from BedMachine v5 (Morlighem et al., 2022). Spin-up geometry from all SMB scenarios can be found in Fig. A1.

where $\overline{T}_{1850-2000}$ and $\overline{P}_{1850-2000}$ are monthly mean temperature and precipitation date from Box (2013), and $\Delta T_t$ and $\Delta P_t$ are the corresponding paleo anomalies from Badgeley et al. (2020). Every model run uses the Holocene climatology from 205    12,500 years ago up until 1850 CE, with the final 150 years using the mean modern values from Box (2013).

## 3.5    Younger Dryas spin-up

In order to obtain an initial ice-sheet state that fits with geological data and is internally consistent, we start a spin-up simulation from the modern day ice surface from BedMachine v5 (Morlighem et al., 2022), and force the model with a constant climate from 12,500 BP, under the assumption that Ryder Glacier was in a stable position during the Younger Dryas (YD). We conduct 210    a spin-up for the four different SMB scenarios, while maintaining consistent boundary conditions of ice thickness, temperature and velocities from the larger continental-scale model (cf. Section 3.1). It takes 20,000 model years until ice volume, temperature, and velocities reach an equilibrium. We do not allow calving during the spin-up and apply an arbitrary *deepmelt* rate of 10 m/yr after the first 2,000 years to allow for grounding line advance. This ocean melt value is chosen as it produces a stable terminus position and while it is significantly lower than contemporary submarine melt rates at Ryder Glacier (Wilson et al., 215    2017), we argue that ocean melt would be subdued owing to a reduction in subglacial discharge in a much colder climate (Fig. 3a).





At the end of the 20,000 year spin-up, Ryder Glacier thickens substantially and advances ∼90 km to reach the mouth of Sherard Osborn Fjord, where the grounding line rests on the outer sill and an ice shelf extends outward towards the Lincoln Sea (Fig. 4). The final spun-up state across the four different SMB scenarios is very consistent (Fig. A1), owing to little climatic variation between the SMB scenarios at the YD (Fig. 3). The position of the grounding line fits with the oldest sediments found on top of till in Sherard Osborn Fjord, dated to $12 \pm 0.5$ ka BP (O'Regan et al., 2021). Furthermore, the ice shelf emanating from the fjord aligns with the hypothesised Lincoln Sea Ice Stream and ice shelf conditions across Northern Greenland (Funder and Larsen, 1982; Dawes, 1986; Larsen et al., 2010). We take these YD states of Ryder Glacier to be the initial state of our Holocene transient runs.

### 3.6 Transient experiments

Our transient experiments are designed to assess how atmospheric and ocean-ice interactions influenced the retreat of Ryder Glacier in the Holocene. We begin with five initial oceanic scenarios that vary the calving threshold (Section 3.2), and ocean *deepmelt* rate (Section 3.3), from modern *Reference* values (Table 1). These five sets of ocean forcings are run across the four different Holocene SMB scenarios previously discussed (Section 3.4).

Our *Reference* calving threshold for floating ice is 300 kPa based on a contemporary study of the neighbouring Petermann glacier by Åkesson et al. (2022), where the threshold was calibrated to reproduce the present day floating tongue. For our *High_Calving* scenario, we lower the threshold to be 200 kPa to implicitly account for an extended calving period during the HTM when sea ice conditions, where much more seasonal (Detlef et al., 2023) as opposed to the almost perennial sea ice in the contemporary fjord. Finally, we set a threshold of 1000 kPa for our *Low_Calving* scenario, the same value as our grounded ice calving threshold, with the idea of preserving Ryder Glacier's floating tongue throughout the Holocene. At the beginning of all transient runs the floating calving threshold is set to 1000 kPa before linearly transitioning to the defined value in the ocean scenario between 11,500 ka BP and 10,000 ka BP as temperatures rise from the YD into the Holocene.

For melt under the floating tongue, we set our *Reference* value for *deepmelt* to be 40 my$^{-1}$. This aligns with oceanic modelling studies of Ryder Glacier as well as satellite observations of Ryder and other north Greenland glaciers (Wilson et al., 2017; Millan et al., 2023; Wiskandt et al., 2023). For a *High_Melt* scenario we set the *deepmelt* rate to be 100 my$^{-1}$, similar to modelled peak melt rates under Ryder Glacier's ice shelf with a warming of AW temperatures by 2°C (Wiskandt et al., 2023). For a *Low_Melt* scenario we take a *deepmelt* rate of 10 my$^{-1}$, the same value used in the YD model spin-up. The transition from a consistent *deepmelt* rate of 10 m/yr during our spin-ups simulations to the defined melt rate for the transient simulation is achieved by linearly increasing the melt rates between 11,500 BP and 10,000 BP.

Furthermore, we run two extreme ocean scenarios, here termed *High* and *Low*, where we combine the respective *deepmelt* rates and floating calving thresholds from their individual high and low scenarios together. These two ocean scenarios are only run with the upper and lower limits of our SMB pathways.





**Table 1.** A summary of different ocean forcings with respective calving thresholds for floating ice and submarine melt values. In bold are the ocean scenarios that are only run with the upper and lower SMB scenarios (Fig.3); high-temperature and low-temperature (High/Low) as well as low-temperature and high-precipitation (Low/High). Note that the spin-up stimulations do not include calving.

| Ocean Scenario | *Deepmelt* (m yr$^{-1}$) | Calving Threshold (kPa) |
| --- | --- | --- |
| Spin-up | 10 | - |
| Reference | 40 | 300 |
| Low Melt | 10 | 300 |
| High Melt | 100 | 300 |
| Low Calving | 40 | 1000 |
| High Calving | 40 | 200 |
| **Low** | 10 | 1000 |
| **High** | 100 | 150 |

## 4 Results

Here we describe the results of our Holocene transient simulations of Ryder Glacier by first focusing on the marine retreat and
ice dynamics of the glacier (Section 4.1), followed by the terrestrial evolution of the ice-sheet margin (Section 4.2).

### 4.1 Marine Holocene evolution

The evolution of Ryder Glacier in our simulations is described in the context of the lithologic units (LU) from a sediment core taken in Sherard Osborn Fjord that can be used for an idealised comparison on the glacier's behaviour (O'Regan et al., 2021).

### 4.1.1 Early Holocene retreat

Ryder Glacier retreats from the outer sill between 10.4 and 9.4 ka BP in all our Holocene simulations (Fig. 5). This is ~2000 years later than indicated by the age of the oldest sediments recovered from the outer sill in Sherard Osborn Fjord, which are dated to $12 \pm 0.5$ ka BP (LU6). The simulated retreat dates instead align well with the basal age of the next lithologic boundary, LU5, dated to $10.7 \pm 0.4$ ka BP, found between both sills in Sherard Osborn Fjord which marks the beginning of sedimentation in the mid-fjord region (Fig. 2a).

There is minimal variation in the timing of retreat from the outer sill across the different ocean scenarios (Fig. 5), with the glacier's geometry very consistent at 10.3 ka (the minimum age of LU5; Fig. 6 a-d). Instead, the timing of retreats varies with the choice of SMB pathways, that is, for high or low scenarios of temperature and precipitation. Simulations using the High/Low (temperature/precipitation) scenario retreating first (10.4 to 10.2 ka BP), followed by Low/Low (10.2 to 9.9 ka BP) and High/High (10 to 9.8 ka BP) and finally Low/High (9.8 to 9.4 ka BP). We consistently observe that this sequence of SMB
scenarios dictates the timing of retreat within each ocean forcing throughout the Holocene (Fig. 5).





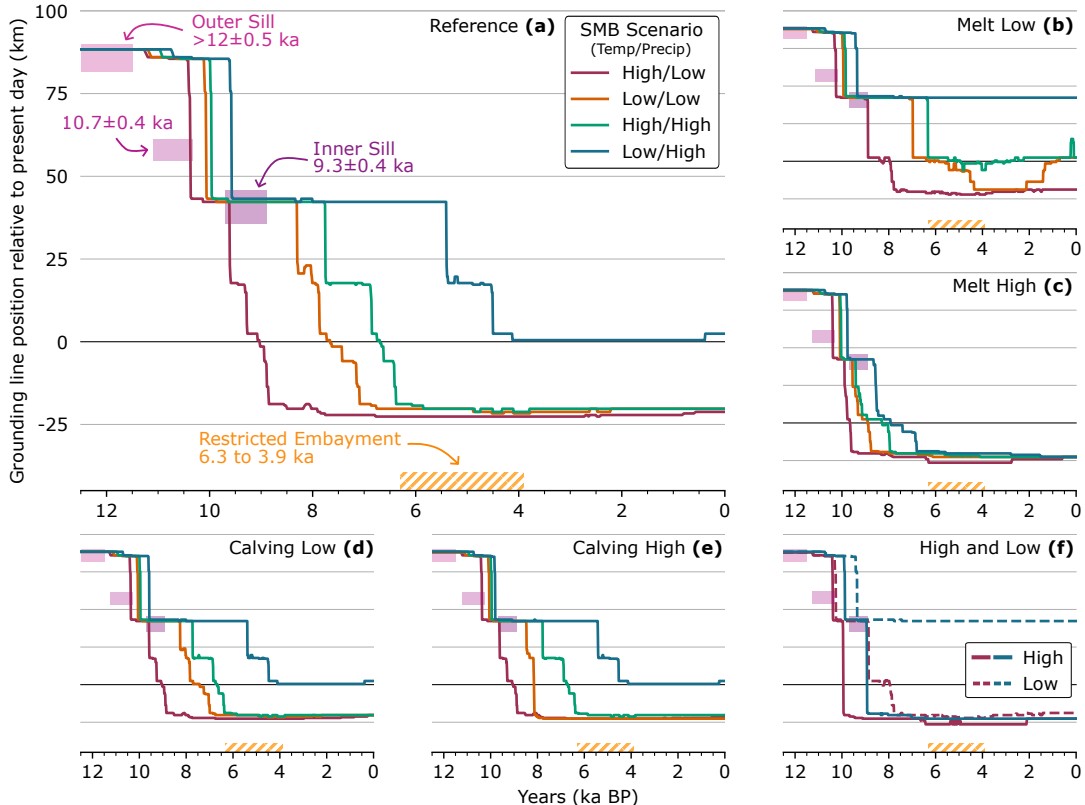

**Figure 5.** Evolution of the grounding line relative to Ryder's present day position for all transient runs. Line colour defines the SMB scenario for the transient runs: High/Low (red), Low/Low (Orange), High/High (Green) and Low/High (Blue). Each subplot represents a different ocean scenario: **(a)** *Reference*, **(b)** *Melt_Low*, **(c)** *Melt_High*, **(d)** *Calving_Low*, **(e)** *Calving_High* and **(f)** both the extreme *High* and *Low* ocean forcings. Shaded boxes highlight the position of sediment cores and their minimum ages, alongside possible timing of retreat into the restricted embayment upstream of the present day grounding line. All dates are sourced from O'Regan et al. (2021).

While the timing of retreat from the outer sill differs between SMB scenarios, the pattern and pace of retreat does not. In all runs, once Ryder begins to detach from the outer sill the grounding line quickly retreats ∼40 km in less than 50 years (Fig. 5), implying a mean retreat rate >800 m y$^{-1}$. Notably, there is no grounding-line standstill in the deeper mid-fjord, and continuous retreat is observed, irrespective of the choice of atmospheric and oceanic forcing, until the grounding line reaches the inner sill.

Here, the grounding line remains stable in all runs for at least 500 years. It is noted here that simulations Low/High: *Melt_Low* and Low/High: *Low* do not retreat further than the inner sill, and remain grounded for the reminder of the Holocene simulation (Fig. 5 b and f).

Our simulations show a wide range of dates for retreat from the inner bathymetric high: 9.7 to 5.4 ka BP (Fig. 5). For runs forced with ocean scenarios: *Reference*, *Calving_Low* and *Calving_High*, retreat ages are consistently between 9.6 and 5.4 ka

BP (Fig. 5a, d and e). The timing of retreat is later for simulations using the *Melt_Low* ocean forcing, between 8.9 and 6.4 ka





**Figure 6.** Evolution of Ryder Glacier in 2D cross section views along the centre flowline during the Holocene at specific time stamps. Dates of 10.3 ka **(a-d)**, 8.9 ka **(e-h)**, 3.9 ka **(i-l)** and 0 ka **(m-p)**, represent specific landmarks in the evolution of Ryder as discussed in O'Regan et al. (2021). Each row represents a different ocean forcing used in the transient simulation (Table 1), from top down this includes: *Reference*, *Melt_High*, *Melt_Low* and both extreme *High* and *Low* Scenarios. Ocean scenarios *Calving_High* and *Calving_Low* are found in the supplementary information as their behaviour does not differ considerably from the *Reference* scenario (Fig. A2). Each plot contains the 4 different SMB scenarios: High/Low (red), Low/Low (Orange), High/High (Green) and Low/High (Blue).




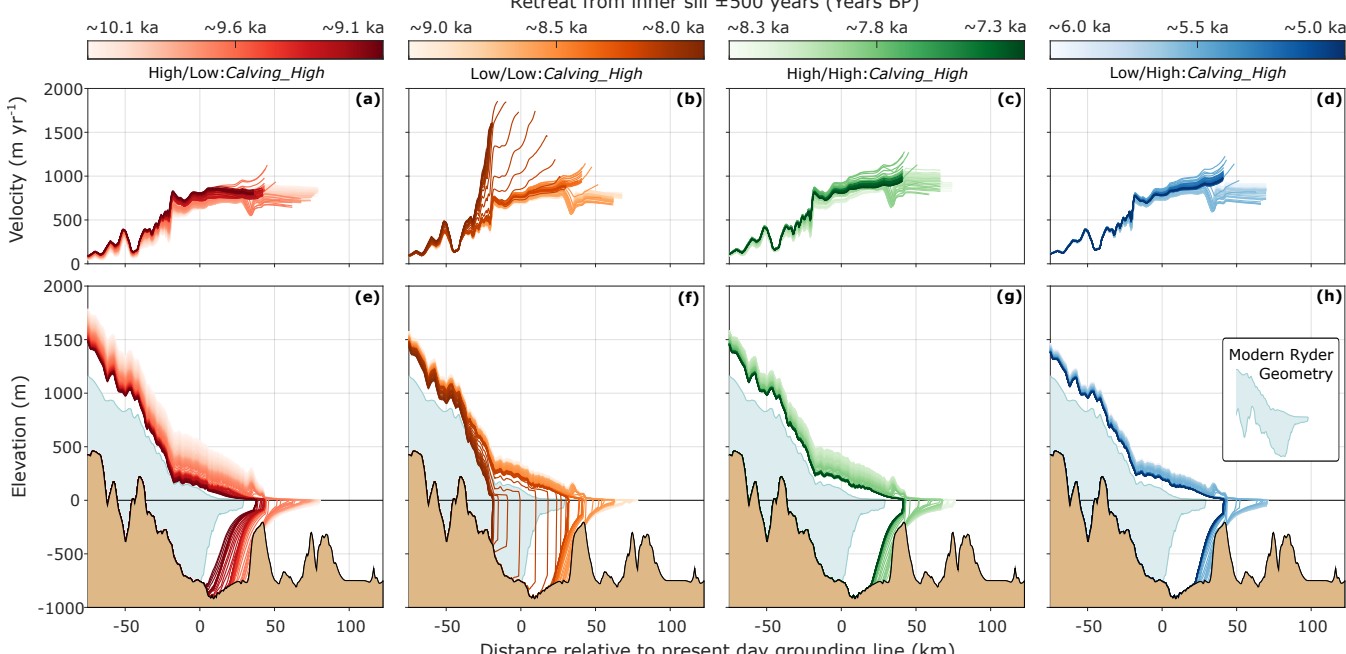

**Figure 7.** Ice dynamics of Ryder Glacier in the 500 years prior to and after retreat from the inner sill in Sherard Osborn Fjord for all transient runs forced with the *Calving_High* ocean scenario. (**a-d**) Changes in velocity along the centre flowline. (**e-h**) Changes in the geometry of Ryder Glacier using 2D cross sections from the centre flowline.

BP, albeit with Low/High: *Melt_Low* remaining on the inner sill for the remainder of the Holocene simulation (Fig. 5b). Dates of retreat from the inner sill are earlier and much more concentrated for runs forced with *Melt_High* and *High* ocean scenarios, at 9.9 to 8.7 ka BP and 10 to 9 ka BP, respectively (Fig. 5 c and f).

It is only in simulations using an elevated ocean melt in *Melt_High* and *High* ocean forcing, as well as all runs forced with
the most extreme SMB of high-temperature and low-precipitation (High/Low), that we see both grounding line retreat and the floating tongue detached from the inner sill that matches the onset of continuous sedimentation at this location: $9.3 \pm 0.4$ ka BP (LU5; Fig. 6e-h). All remaining simulations remain grounded on the inner sill until at least ~8.3 ka BP (Fig 5).

### 4.1.2 Holocene minimum and ice tongue collapse

Ryder Glacier's Holocene minimum extent is captured by the sediment unit LU3 found throughout sediment cores taken in
Sherard Osborn Fjord and spanning $6.3 \pm 0.3$ to $3.9 \pm 0.4$ ka BP. This unit marks a period of low sedimentation and the presence of IRD, interpreted by O'Regan et al. (2021) as the collapse of the ice tongue and the retreat of the glacier into the restricted embayment ~40 km upstream of the present grounding-line position (Fig. 2a).

None of our transient Holocene simulations match, in detail, this hypothesised retreat to the restricted embayment (Fig. 6). We simulate a maximum retreat of Ryder Glacier to just in front of the restricted embayment, ~25 km behind the present day



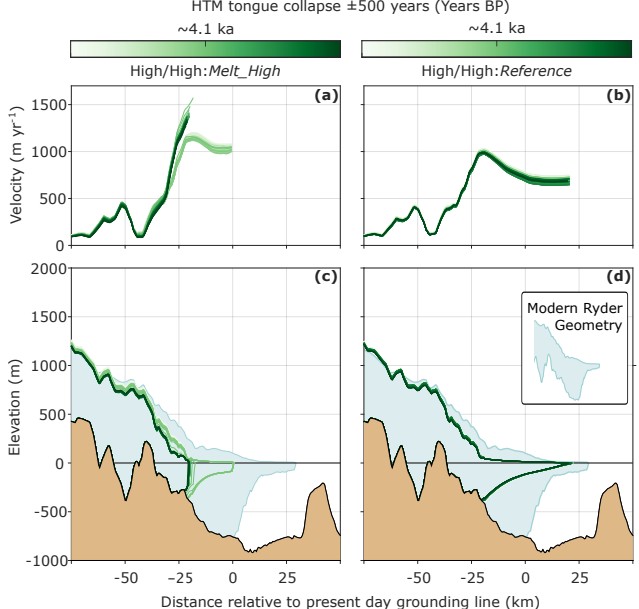

**Figure 8.** Ice dynamics in the 500 years prior to and after the loss of the floating tongue during the transient run High/High:*Melt_High* and ice dynamics of the High/High:*Reference* during the same period. (**a-b**) Change in velocity along the centre flowline. (**c-d**) Change in the geometry of Ryder Glacier using a 2D cross section from the centre flowline.

grounding-line position (Fig. 5). A retreat to this position occurs in all simulations using the *Reference* ocean forcing with the exception of the run using the mildest SMB scenario Low/High (Fig. 5a). Timing of retreat again follows the same pattern of SMB; with a Holocene minimum occurring at 7.9 ka BP for SMB High/Low, 7.1 ka BP for Low/Low, 6.4 ka BP for High/High ka BP and 4.5 ka BP for Low/High.

The same timing and extent of retreat is also seen in the *Calving_High* and *Calving_Low* scenarios, with the exception
of run Low/Low: *Calving_High* that retreats ∼900 years earlier (8 ka BP) than its *Reference* run due to the collapse of the floating tongue (Fig. 7). For *Melt_Low* simulations, a similar Holocene minimum position of ∼25 km behind the present-day grounding line is reached with High/Low and Low/Low SMB scenarios, albeit later than their *Reference* ocean simulations, while High/High: *Melt_Low* does not retreat past the present-day grounding line (Fig. 5 a and b). All runs using the *Melt_High* ocean forcing retreat to this common Holocene minimum position, but again retreat is found to be both faster and more
concentrated, with a minimum extent of Ryder reached between 9.5 and 6.5 ka BP.

We find Ryder's floating tongue collapses during our Holocene simulations through two distinct manners. The first occurs when Ryder retreats from the inner sill (c. at km 40 in Fig. 7). This happens in both runs using the *High* ocean forcing as well as Low/Low: *Calving_High*. Figure 7 compares the retreat of all four *Calving_High* simulations for the 500 years before and after the grounding line detaches from the inner sill. Runs forced with SMB scenarios: High/Low, High/High and Low/High
follow the same pattern of behaviour, albeit the timing of retreat differs. Therein, Ryder's retreat from the inner sill is followed





by a steady grounding-line retreat of between 20 and 30 km in the following 500 years (Fig. 7 e, g and h). Terminus velocities of the glacier prior to retreat from the sill are $\sim 750$ m yr$^{-1}$, as the grounding line moves from the sill they increase to >1,000 m yr$^{-1}$, before gradually subsiding to velocities below 1,000 m yr$^{-1}$. Run Low/Low: *Calving_High* initially follows the same behaviour, however large calving rates which remove the majority of the floating tongue coincides with an acceleration of

terminus velocities to $\sim 1900$ m yr$^{-1}$ and the glacier retreats as a shear cliff face back 35 km in $\sim 40$ years (Fig. 7 b and f). The glacier then rests at its Holocene minimum as a grounded tidewater glacier, with terminus velocities of $\sim 1600$ m yr$^{-1}$.

The second mechanism for tongue collapse occurs in all runs forced with the *Melt_High* ocean scenario. The behaviour is exemplified in Figure 8 by run High/High: *Melt_High* where the glacier, with the grounding line already at its Holocene minimum sees its 25 km tongue disintegrate and terminus velocities accelerate from $\sim 1,000$ m yr$^{-1}$ to $\sim 1,500$ m yr$^{-1}$. No

further grounding-line retreat is observed, as the glacier remains in shallow waters in front of the restricted embayment. The timing of collapse varies depending on SMB scenarios: High/Low at 7.5 ka BP, Low/Low at 6.2 ka BP, High/High at 4.1 ka BP and finally Low/High at 2.1 ka BP. The collapse behaviour is contrasted by the use of the *Reference* ocean forcing in High/High: *Reference* (Fig 8b and d), where a floating tongue of almost 50 km in length is sustained throughout the Holocene and velocities at the terminus remain consistently between 600 and 700 m yr$^{-1}$.

### 4.1.3 Neoglacial regrowth

The final lithologic units found in Sherard Osborn fjord, beginning at $3.9 \pm 0.4$ ka BP, are characterised by increase sedimentation rates and the deposition of laminated sediments, marking the advance of the grounding line and the eventual regrowth of Ryder's ice tongue towards it LIA maximum (Fig. 2b).

In our transient simulations, we only find a late Holocene re-advance of the grounding line using the *Melt_Low* or extreme

*Low* ocean forcing (Fig. 5b and f). Therein, the most pronounced readvance occurs in the Low/Low: *Melt_Low* run where the grounding line migrates $\sim 25$ km. The re-advance is more subtle in runs High/Low: *Melt_Low* and High/High:*Melt_Low* at $\sim 10$ km, although substantial thickening upstream of the grounding is observed (Fig. 6k and o). This same pattern of thickening is observed in runs Low/High:*Melt_Low* and Low/High:*Low*, although the grounding line, which has not retreated from the inner sill during the HTM, does not show any advance. In all aforementioned runs, the floating tongue advances into Sherard

Osborn Fjord at a similar rate to grounding line migration. In runs where there are no perceived grounding-line advances, such as those using the *Reference* ocean scenario (Fig. 6i and m), there is no increase in length of the floating tongue towards the LIA maximum. Furthermore, there is no regrowth of the floating tongue in runs where a collapse occurs in the mid-Holocene (Fig. 6j and n).

## 4.2 Terrestrial Holocene evolution

Here we explore the evolution of the terrestrial ice-sheet margin during our simulations in the context of reconstructed ice-sheet isochrones found in the PaleoGrIS database (Leger et al., 2023). Firstly, we assess the simulated retreat of the ice-sheet margin during the early Holocene from 10 ka to 8 ka BP (Section 4.2.1), over which time our model domain transitions from nearly entirely ice covered to roughly the same ice coverage as found at present-day (Fig. A3). 8 ka BP also marks a pause in ice





retreat found across all our simulations, likely relating to the Warming Land Stade inferred by Kelly and Bennike (1992) (Fig.
A3). Furthermore, we describe the neoglacial advance of the ice-sheet margin by comparing a minimum ice extent, taken at
6 ka BP for all runs except those forced with SMB scenario Low/High when the minimum occurs later at 4 ka BP (Fig. A3),
with the final state of our simulation representing present day (Section 4.2.2).

### 4.2.1   Warming Stade retreat

The terrestrial ice margin's rate of retreat varies considerably with the choice of SMB scenario (Fig. 9). The most negative SMB
scenario, comprised of high-temperature and low-precipitation (High/Low), produces an ice margin that is consistently 30 -
40 km behind the reconstructed margin from the geological archive (Fig. 9a, e and i). The ice margins from simulations using
the Low/Low (Fig. 9b, f and j) and High/High (Fig. 9c, g and k) SMB scenarios follow a similar retreat rate during the early
Holocene, yet again exceed the rate of retreat in the reconstructed isochrones across Warming Land but more closely match the
retreat around Steensby Glacier and St. George Fjord. Finally, simulations using the least negative SMB scenario, containing
low-temperature and high-precipitation (Low/High), produce a retreat that tracks the reconstructed isochrone especially well
at 9 and 8 ka BP, despite producing a slighter slower retreat at 10 ka BP (Fig. 9 d, h and l).

While the reconstructed ice margins from the PaleoGrIS database provide valuable insights into the Holocene extent of
GrIS, we note that across the region of Northern Greenland the isochrones are rated with *low* confidence, the third lowest of
four confidence levels discussed in Leger et al. (2023). This is due to the sparse number of radiocarbon ages and the almost
complete lack of exposure event ages across the region. Nevertheless, the series of moraines that have been dated in Northern
Greenland show a standstill in Holocene retreat between 9 and 8 ka BP and indicate that ice sheet had not yet retreated behind
its modern day extent (Fig. 1; Kelly and Bennike, 1992). This underscores that the retreat of the ice margin in the most negative
SMB scenario High/Low is too pronounced, with simulated ice margin already behind the present day ice sheet by 9 ka BP.

### 4.2.2   Holocene minimum and regrowth

The minimum glaciated extent of the model domain during the transient Holocene runs is shown in Figure 10a-d. The terres-
trial margin, on both Warming Land and Wulff Land, has retreated behind that of the present-day ice sheet in all simulations.
There is little variation in the ice margin across the different ocean scenarios used for runs forced with Low/Low, High/High
and Low/High SMB scenarios, with the simulated ice-sheet minimum extent sitting between 10 and 20 km behind the contem-
porary ice margin on Warming Land. For simulations using the High/Low SMB, modelled retreat is greater, with the margin
retreated at least 30 km behind the present-day ice sheet and Steensby Glacier withdrawing from St. George Fjord. Furthermore,
simulation High/Low: *Melt_High* produces a retreat that is 10 km further in land than all other ocean forcings.

Regrowth of the terrestrial margin towards its present-day location occurs in all our simulations with the exception of those
using the *Melt_High* ocean scenario (Fig. 10e-h). For such simulations, the ice-sheet margin remains almost unchanged from
its Holocene minimum position, 20 - 40 km behind the present ice sheet depending on the chosen SMB scenario. All remaining
runs using the High/High and Low/High SMB scenarios, both containing the high-precipitation pathway, end with an ice-sheet
margin that is very consistent with the present-day ice sheet both across Warming Land and Wulff Land. For using SMB



**Figure 9.** Evolution of the ice sheet margin across the model domain at 10 **(a-d)**, 9 **(e-h)** and 8 **(i-l)** ka BP. Each column shows the set simulations forced with each of the various SMB scenarios, while coloured ice sheet margins depict the ocean scenario using in the simulation: *Reference*: navy, *Melt_Low*: light blue, *Calving_Low*: blue, *Melt_High*: light orange and *Calving_High*: dark orange. Purple lines represent the reconstructed ice sheet margins from the PaleoGrIS database (Leger et al., 2023).

scenarios High/Low and Low/Low there exists variation in the final position of the ice front depending on the chosen ocean forcing, however all simulations produced an final ice margin that is within 10 km of the contemporary ice sheet.



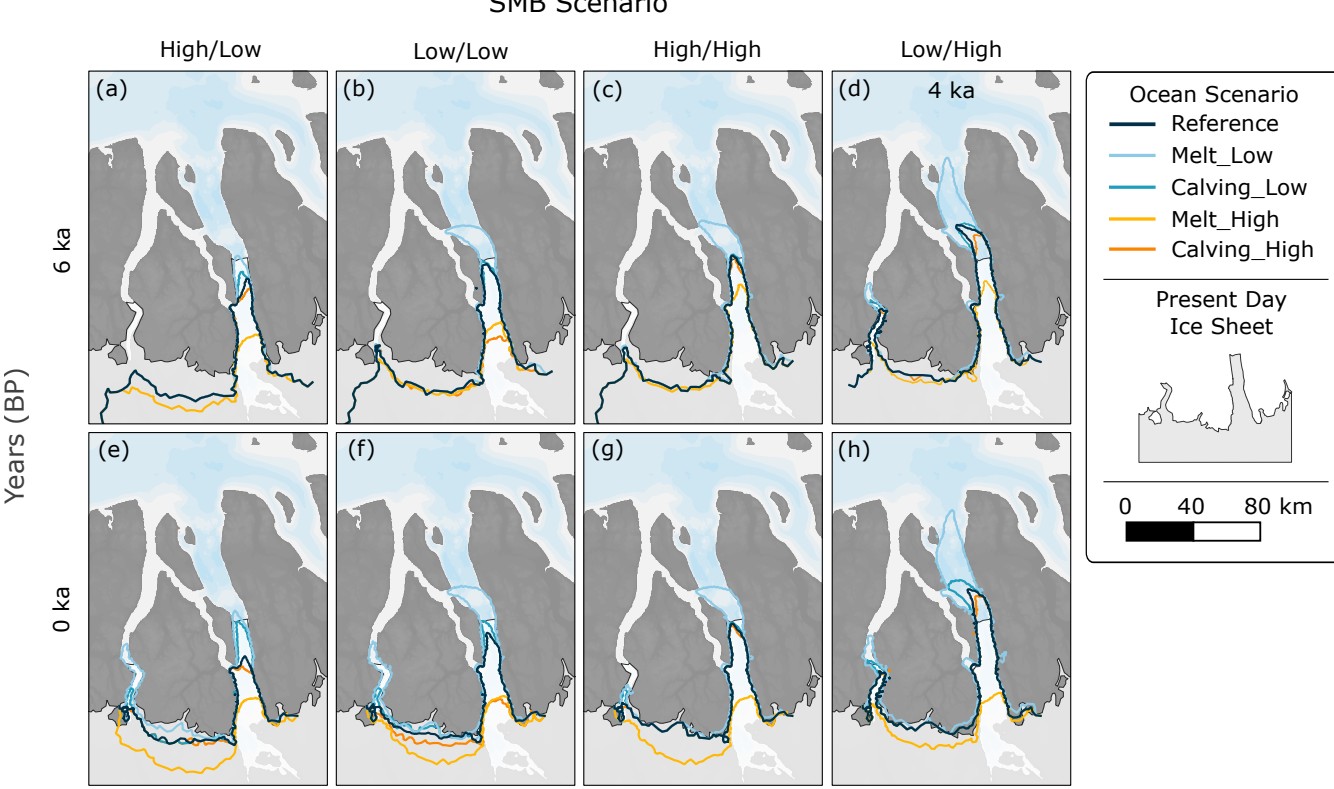

**Figure 10.** Comparison of the ice sheet margin at the Holocene minimum (**a-d**), and the end of our simulations representing present day (**e-h**). Each column shows the set simulations forced with each of the various SMB scenarios, while coloured ice sheet margins depict the ocean scenario using in the simulation: *Reference*: navy, *Melt_Low*: light blue, *Calving_Low*: blue, *Melt_High*: light orange and *Calving_High*: dark orange.

## 5 Discussion

We have simulated the Holocene evolution of Ryder Glacier across different plausible atmospheric- and oceanic-forcing scenarios with the glacier's terminus position and ice dynamics constrained by marine and terrestrial proxy records. In summary, we find that variations in SMB have a distinct impact on timings of retreat for both terrestrial and marine ice-sheet margins. The climate pathway built from low-temperature and high-precipitation (Low/High), matches the early-Holocene terrestrial retreat across our domain best, in contrast to the most extreme scenario of high-temperature and low-precipitation (High/Low), which

produces retreat that far exceeds the rate found in the terrestrial record (Fig. 9). For the marine margin of Ryder, enhanced ocean melt plays a key role in driving grounding-line retreat to match the sediment archive in all but the most extreme SMB scenario (Fig. 5). Moreover, ocean melt is crucial to initiate the collapse of Ryder's ice tongue in the HTM (Fig. 8), while the readvance of the grounding line towards its modern day position only occurs with subdued ocean melt forcing (Fig. 5b). Furthermore, the variations in the calving threshold have a minimal impact during the early Holocene retreat, likely due to





the continued presence of the floating tongue with its stability aided by the narrowing of Sherard Osborn Fjord. The major components influencing the Ryder Glacier's behaviour in our simulations will be discussed in more detail below.

## 5.1 Holocene warming and the importance of precipitation

Atmospheric changes, associated with the pronounced shifts in surface temperatures and precipitation from the YD into the Holocene, exerted a first-order control on the behaviour of Ryder Glacier in our simulations. This is clear from the initial
homogeneous retreat, >10 ka BP, found across all simulations in response to an abrupt rise in temperature and independent from the choice of ocean forcing. Subsequent variations in temperature and precipitation invoke heterogenous retreat rates during the Holocene, yet the choice of SMB scenario provides a consistent pattern of retreat across all ocean forcings; that ranges from the most extreme SMB scenario of high-temperature and low-precipitation to the mildest scenario of low-temperature and high-precipitation. We find this mildest SMB scenario to produce a terrestrial ice margin that fits best with reconstructed
ice sheet positions in the geologic records (Fig. 9). This is in contrast to the findings of Cuzzone et al. (2022), who used the same climatic pathways for a Holocene simulation of Kangiata Nunaata Sermia glacier, Southwestern Greenland. Furthermore, precipitation exerts a substantial control in modulating mass loss during the early Holocene in our simulations, so much so that SMB scenarios using the low-precipitation pathways produce the fastest retreat rates of both the grounding line and terrestrial ice margins (Fig. 5 and 9). This implies that, when the catchment of Ryder Glacier was starved of snowfall, grounding-line
retreat occurred. These results again contrast with those from South Western Greenland, where variations in temperature appear to have played a greater role in determining the rate of retreat (Briner et al., 2020; Cuzzone et al., 2022).

We attribute the strong impact of precipitation during the Holocene to the environmental characteristics of Northern Greenland. The region, in its contemporary setting, experiences some of the lowest precipitation rates found on the ice sheet (Box et al., 2013; McIlhattan et al., 2020), owing to both its distance from major open water bodies, as the nearby Arctic Ocean is
sea-ice covered, and to cooler temperatures at extreme latitudes. Consequently, Northern Greenland is especially vulnerable to rapid atmospheric warming (Noël et al., 2019). Reconstructed early Holocene precipitation suggests lower than modern rates of accumulation (Fig. 3b), alluding to the North Greenland Ice Sheet being highly sensitive to early Holocene warming. Analyses of biomarkers in sediment cores from the Lincoln Sea suggest that the period between about 11.3 and 9.7 ka BP may have experienced seasonal sea ice, implying nearby open water conditions during summers (Detlef et al., 2023). However, it is
not known whether such conditions in the Lincoln Sea directly affected precipitation over the North Greenland Ice Sheet. The sediment core study further suggests a maintained perennial sea-ice cover in the southern Lincoln Sea throughout the reminder of the Holocene. In contrast, towards the mid-Holocene, reduced sea-ice extent in the Baffin Bay and Labrador Sea together with warmer temperatures are thought to have brought greater than modern day levels of accumulation to Northern Greenland (Thomas et al., 2016). Such reasoning has been invoked to explain readvances of valley glaciers and the sustained presence of
local ice caps across Northern and North eastern Greenland during the HTM (Möller et al., 2010; Larsen et al., 2016, 2019), while simulations of Hans Tausen Iskapp showed that precipitation rates higher than modern-day values were crucial in buffering higher temperatures during the HTM and the survival of the ice cap (Zekollari et al., 2017). Greater than modern-day



precipitation rates are only found in our high-precipitation scenario (Fig 3b), and when coupled with a cold pathway produced the most accurate retreat of terrestrial margin during the early Holocene (Fig. 9).

## 5.2 Submarine melt drives inner fjord retreat

Whilst a negative trend in SMB is the overall dominant driver of Ryder Glacier's Holocene retreat, our results indicate that ice-ocean interactions played an increasingly important role as the interglacial progressed. The ocean melt rate is particularly important for the grounding line's retreat from the inner sill in Sherard Osborn Fjord, where our simulations produce a wide range of retreat dates spanning from 9.4 to 5.4 ka BP (Fig. 5). The study of O'Regan et al. (2021) shows that sedimentation on the inner sill, and thus the beginning of grounding line retreat, began at $9.3 \pm 0.4$ ka BP. We find that using a modern *Reference* ocean melt rate, our simulations are only able to match this timing of retreat when using the most negative SMB scenario (High/Low: *Reference*, Fig. 5a and 6e). Such an extreme SMB scenario invokes a terrestrial retreat that is significantly faster than that found in reconstructed isochrones (Fig. 9). Other, more moderate, SMB scenarios, result in grounding-line evolutions lagging behind the sediment record when using *Reference* ocean melt forcings, with retreat from the inner sill not beginning until 8.3 ka BP (Fig. 5a and 6e). It is only when using a *High_Melt* ocean parameterisation that all SMB scenarios yield grounding-line retreats that are in line with the marine record (Fig. 5c and 6f).

The inner-sill has a ~6.2 km wide central zone with a modern depth of ~300 m (Jakobsson et al., 2020). Ocean melt rates at the grounding line atop the inner sill in our melt parametrisation would peak at 26.6 m yr$^{-1}$ for the *Reference* ocean scenario and 66.6 m yr$^{-1}$ for the *High_Melt* simulations, with the latter resembling and even exceeding the modern melt rates seen under contemporary Greenlandic floating tongues with their grounding lines in much deeper waters (Wilson et al., 2017; Millan et al., 2023). While the depth of the modern inner sill impedes the flow of warm AW (Jakobsson et al., 2020), relative sea-level (RSL) estimates from ~8 ka BP indicate the sill to be between 40 and 80 m deeper than present (Lecavalier et al., 2014; Glueder et al., 2022), likely allowing AW to reach the paleo grounding line of Ryder Glacier atop the sill, and drive higher melt rates that otherwise would not be possible in shallower, cooler waters. Cronin et al. (2022) discusses the evidence for the existence of such warm waters in the Lincoln Sea at the mouth of Sherard Osborn Fjord throughout the entire Holocene from a benthic ostracode and foraminifera assemblage. In particular, they point to a pronounced AW signal between 8.5 and 6 ka, that is close in time to the beginning of sedimentation on the inner sill. This stresses that the presence of warm AW in the fjord may have been crucial for the retreat of the grounding line from the sill. The presence of such warm waters is well documented around Greenland during the Holocene (Jennings et al., 2006; Pados-Dibattista et al., 2022), and has been indirectly linked to the collapse of the neighbouring Petermann Glacier's ice tongue during the HTM (Reilly et al., 2019). Furthermore, a similar Holocene modelling study of Jakobshavn Isbræ also found that a pronounced increase in ocean temperatures, and thus ocean melt rate, was required to force the glacier from a stable position atop a bathymetric sill and initiate Holocene retreat into the HTM (Kajanto et al., 2020).

Ocean melt rates are also found to be crucial to the stability of Ryder's floating tongue during the HTM. Our simulations produce a complete break-up of the ice tongue in two distinct manners: first during the early Holocene as the glacier retreats from the inner sill into deeper waters (Fig 7), second, during the mid-Holocene once the grounding line has already retreated





into the shallower waters upstream of the present grounding line (Fig 8). We find the latter, occurring in all simulations using a *High_Melt* ocean forcing, best reflecting the marine record in Sherard Osborn Fjord. The sediment facies that depict the collapse of Ryder's floating tongue, LU3 (6.3 ± 0.3 to 3.9 ± 0.4 ka BP), is dated to occur ∼3 ka years after Ryder's retreat from the inner sill, LU5 (9.3 ± 0.4 to 7.6 ± 0.40 ka BP). The presence of LU4 (7.6 ± 0.40 to 6.3 ± 0.3 ka BP), a faintly laminated facies that lacks evidence of bioturbation, suggests that Ryder maintained its floating tongue for a substantial period of time as the grounding line retreated away from the inner sill (O'Regan et al., 2021). We therefore argue that the collapse of Ryder's tongue is best captured in runs using the *High_Melt* scenario, wherein Ryder retreats steadily through the over-deepened part of the fjord behind the inner sill to its Holocene minimum position. In this scenario, elevated melt rates result in a shortened floating tongue that weakens the glacier's ability to buttress interior ice, thus driving an increase in ice velocities and calving until its eventual collapse (Fig. 8).

### 5.3 Ryder's floating tongue was pivotal in its Holocene evolution

Mass loss through calving has played a key role in the recent evolution of the GrIS and is projected to play a vital role in the coming century (Goelzer et al., 2020; Choi et al., 2021). Yet, such processes are often not explicitly included in paleo-modelling efforts owing to the computational costs associated with high-resolution grids needed to accurately simulate fast flowing ice streams and their associated calving dynamics. Here, we build on the findings of Cuzzone et al. (2022), who included a similar calving setup when simulating the Holocene retreat of Kangiata Nunaata Sermia, South Western Greenland. Cuzzone et al. (2022) note how the inclusion of calving had little impact on the retreating terrestrial ice margin but instead allowed the adjacent marine margins to persist longer in the fjord owing to a greater transport of mass from the interior, which in turn led to thinning in the upper regions of the domain. We find such consistencies across our simulations during the early Holocene, with fast retreat of terrestrial ice across our domain independent of the chosen calving thresholds and ocean melt forcings, while the marine margin of Ryder Glacier is able to maintain its extended position in the fjord despite adjacent retreat on land (Fig. 9).

In our simulations, variations in calving thresholds have little impact on the grounding line evolution and inland thinning of the glacier (Fig. 5a, d and e; and Fig. A2). We attribute this to the sustained ice-tongue presence during the Holocene retreat that suppresses the rate of calving at Ryder Glacier. In the event of ice tongue collapse, either during retreat from the inner sill (Fig. 7) or at the Holocene minimum position (Fig. 8), the loss of buttressing results in an increase in ice discharge and in land thinning. We note that the presence of the ice tongue was particularly important during Ryder's retreat from the inner sill as the grounding line migrated along a retrograde slope that deepened by ∼500 m, where retreat as calving cliff face produced exaggerated retreat (Fig. 7). The overdeepening behind the inner sill coincides with a narrowing of Sherard Osborn Fjord to 10 km. This bottleneck continues for 40 km and has been linked to the stability of the contemporary ice tongue and calving front (Holmes et al., 2021), where narrowing fjord walls increase lateral drag and aide in modulating grounding line retreat and, crucially, floating tongue stability (Åkesson et al., 2018b; Frank et al., 2022).

Our simulations indicate the Ryder's floating tongue was present during the entire >4000 year retreat from the outer sill until its collapse at the glacier Holocene minimum position, 6.3 ± 0.3 ka BP. The geometry of Sherard Osborn Fjord was crucial in



maintaining the stability of the tongue over such protracted timescales. This allowed Ryder's ice tongue to outlast the floating extension at the neighbouring Petermann Glacier, which collapsed at 6.9 ka BP only ~600 years into the glacier's retreat from the outer sill in Petermann Fjord after which the fjord's geometry begins to deepen and, crucially, widen (Reilly et al., 2019). With the collapse of Nioghalvfjerdsbræ (79N) occurring even earlier, by 8.5 ka BP (Smith et al., 2023), its very likely that

Sherard Osborn Fjord contained the last major floating extension from the Greenland Ice Sheet during the deglaciation from the LGM.

     The heightened discharge from a collapse of the floating ice tongue at Ryder is found to restrict any re-advance of both the terrestrial and marine margins during the late Holocene in our simulations (Fig. 6n and 10). This implies that a shift in calving dynamics, driven by the regrowth of ice tongue and increased buttressing, was crucial in aiding the re-advance of Ryder glacier

during the late Holocene. While our experimental set-up does not account for transiently evolving ocean forcings, the lack of ice-tongue regrowth across all SMB scenarios indicates that cooling atmospheric temperatures alone are not sufficient for the regrowth of the floating terminus, and that a reduction in ocean melt rate and/or a reduction in calving owing to the formation of a rigid ice mélange was required. This aligns with the results of Åkesson et al. (2022) who found that a decrease in both ocean melt and calving was required to regrow Petermann Glacier's ice tongue in the event of any collapse as well as work by

Kajanto et al. (2024), where a decrease in calving rate was required re-form the ice tongue of Jakobshavn Isbræ and readvance the grounding line to the glacier's Little Ice Age position.

     Our findings emphasise the role of floating tongues in controlling ice discharge over centennial and millennial time scales and stresses the importance of both including floating ice and calving dynamics when undertaking paleo-modelling studies of the GrIS. This is especially pertinent to Northern Greenland; where ice streams at the LGM were thought to have coalesced

and formed ice shelf conditions into the Lincoln Sea (Dawes, 1986; Larsen et al., 2010) before collapsing during the HTM and then regrowing in the late Holocene to exert an important buttressing force on ice flow from the GrIS interior. Continental-scale model reconstructions of the GrIS during the Holocene which omit the inclusion of floating ice and calving, or do not have the required model resolution to capture grounding line dynamics in narrow Greenlandic fjords, may lack the required fidelity to accurately simulate the ice-sheet's evolution (Simpson et al., 2009; Lecavalier et al., 2014).

While we have discussed the importance of fjord geometry and bathymetry to the evolution of Ryder Glacier, we acknowledge one of the limitations of our experimental set-up is lack of transient RSL changes during the Holocene. Despite this, we tentatively assume that including such dynamics would not have a significant impact on our model results, taking note from a similar Holocene study of Jakobshavn Isbræ by Kajanto et al. (2020), which found RSL changes had minimal impact on the in-fjord retreat as such variations are small compared to fjord depth. While the coalescing of the Greenland and Innuitian

ice sheets brought RSL ~80 m higher at the beginning of the Holocene (Lecavalier et al., 2014; Glueder et al., 2022), the pronounced depths of Sherard Osborn Fjord mean it is likely that any influence would have been restricted to the shallow inner sill as discussed in Section 5.2. Future work, which will aim to simulate the Holocene deglaciation of the entire northern sector of the GrIS, will include RSL variations either through new solid earth feedbacks currently being implemented into ISSM or through prescribed forcings of sea level changes from prior research (e.g. Caron et al., 2018).





## 5.4 Ryder Glacier in the future

The retreat of Ryder to its HTM minimum provides a crucial analogue to understand how the glacier will evolve when confronted with sustained temperatures above pre-industrial levels. Our simulations indicate that both atmospheric- and ocean-ice interactions will dictate Ryder's future behaviour. At present, Ryder's floating tongue and grounding line sits within the bottleneck of Sherard Osborn Fjord, providing significant buttressing and stability. The calving front has remained unmoved for half a century, owing to fjord geometry (Holmes et al., 2021). If the ice tongue at Ryder persists, the grounding line will undergo a steady retreat along the prograde topography inland of its current position, with the speed of retreat a function of both SMB and submarine melt, with the later able to weaken to floating tongue's ability to buttress leading to enhanced discharge (Fig. 8). Despite being shielded from warm AW by the shallower inner sill (Jakobsson et al., 2020), melt rates at the grounding line of Ryder have increased over recent decades while the floating tongue has thinned (Millan et al., 2023). Submarine melt is not just a product of ocean heat but also turbulent processes at the ice-ocean interface that are linked to the upwelling of buoyant plumes fed by the discharge of fresh water at the grounding line (Slater and Straneo, 2022). At Nioghalvfjerdsbræ(79N), Greenland's largest floating terminus that is also shielded by a warm AW by bathymetric high (Hill et al., 2017; An et al., 2021), extreme melt rates of 100 m yr$^{-1}$ have been linked to increased subglacial discharge from enhanced surface melt (Zeising et al., 2024). Rising melt rates at Ryder have coincided with a substantial rise in supraglacial lakes on the glacier (Otto et al., 2022), the build-up and drainage of which are likely to deliver an increased flux of fresh water to the grounding line that aid in transferring heat from ocean to ice. Therefore, despite the protection offered by the inner-sill in Sherard Osborn Fjord, ocean melt is set play a crucial in the grounding line retreat and tongue stability of Ryder Glacier into the future.

During the HTM Ryder retreated inland of its present position. The most common Holocene minimum position achieved in our simulations is ~25 km behind the modern-day grounding line, in relative shallow waters (~300 m deep) and in front of the restricted embayment (Fig. 6 and 8). The ever decreasing marine ice thickness heightens the role of atmospheric-ice interaction and indicates that further retreat would require a greater negative trend in SMB. This is best exemplified in both runs that use the extreme *High* ocean forcing which retreat to their Holocene marine minimum by 9 ka BP (Fig. 6h), only to remain unmoved during the HTM. While we are unable to match the grounding-line retreat into the restricted embayment hypothesised by O'Regan et al. (2021), the inherent uncertainties in SMB reconstructions over the Holocene mean that further retreat from our simulated position cannot be ruled out in HTM like climate. Yet, owing to the substantial thickness of simulated inland ice, we believe that it is unlikely that Ryder retreated significantly further and became entirely terrestrial during the Holocene. Instead, it appears that Ryder remained a marine terminating glacier during the HTM, with the grounding line in shallow waters in front of, or in, the restricted embayment. This implies that under a future climate the resembles that of mid-Holocene, Ryder will retain its connection to the ocean and will continue to contribute to global sea level rise through both runoff and discharge.

## 6 Conclusions

Geological evidence from terrestrial and marine settings can provide valuable insights into the evolution of ice sheets across glacial/interglacial cycles at snapshots in time. Coupling such knowledge with ice-sheet models allows for continuous insight

into the long-term evolution of ice sheets. Here, we employed a high-resolution 3D thermomechanical ice sheet model to explore the evolution of Ryder Glacier throughout the Holocene. By focusing on a single outlet glacier from the GrIS, we are able to run our model at with mesh resolutions under <1 km to both accurately simulate grounding line dynamics and disentangle the influence of ice-ocean and ice-atmosphere interactions during Ryder Glacier's evolution. We find that:

– Rapid retreat of the ice margin in the early Holocene was driven by substantial climatic warming. The retreat of both the terrestrial and marine ice margins are independent of prescribed ocean forcing during the transition out of the YD until ∼10 ka BP.

– The terrestrial ice margin remains insensitive to the choice of submarine melt or calving threshold throughout its Holocene retreat, with the position of the ice margin a function of the prescribed SMB. There-in, rates of precipitation play a key role in modulating both the pace and magnitude of retreat, likely reflecting the arid nature of Northern Greenland.

– After ∼10 ka BP, submarine melt begins to influence the retreat of marine margin of Ryder Glacier in all but the most negative SMB scenario. Heightened submarine melt rates were crucial in retreating Ryder from a stable position on the inner sill as well as inducing the collapse of the ice tongue during the HTM.

– Before its collapse, the paleo-ice tongue at Ryder Glacier was crucial in modulating ice loss during the early Holocene, particularly during Ryder's retreat from the inner sill into a large overdeepening in Sherard Osborn Fjord. We link the ice tongue's stability and influence to the 10 km wide bottleneck in Sherard Osborn, where the contemporary ice tongue and grounding line currently lie.

– Re-growth of the ice tongue was crucial to the readvance of both the terrestrial and marine ice margins during the late Holocene cooling. Changes in SMB alone cannot explain to the re-formation of the tongue, indicating that cooling ocean temperatures and reduced calving rates were required.

While changes in SMB may dictate large-scale ebbs and flows in ice-sheet margins, ice-ocean interactions can play a significant role in the evolution of marine outlet glaciers, and thus discharge from ice sheets, across extended timescales of centuries and millennia. By running Holocene simulations of Ryder Glacier we provide vital analogues for the future evolution of the glacier in a warming climate, stressing both the continued importance of the ice tongue in buttressing ice flow and the role of submarine melt in weakening the ice tongue and driving grounding line retreat

*Code availability.* All the code necessary for running simulations using ISSM is freely available at https://issm.jpl.nasa.gov/download/ .

## Appendix A

### A1



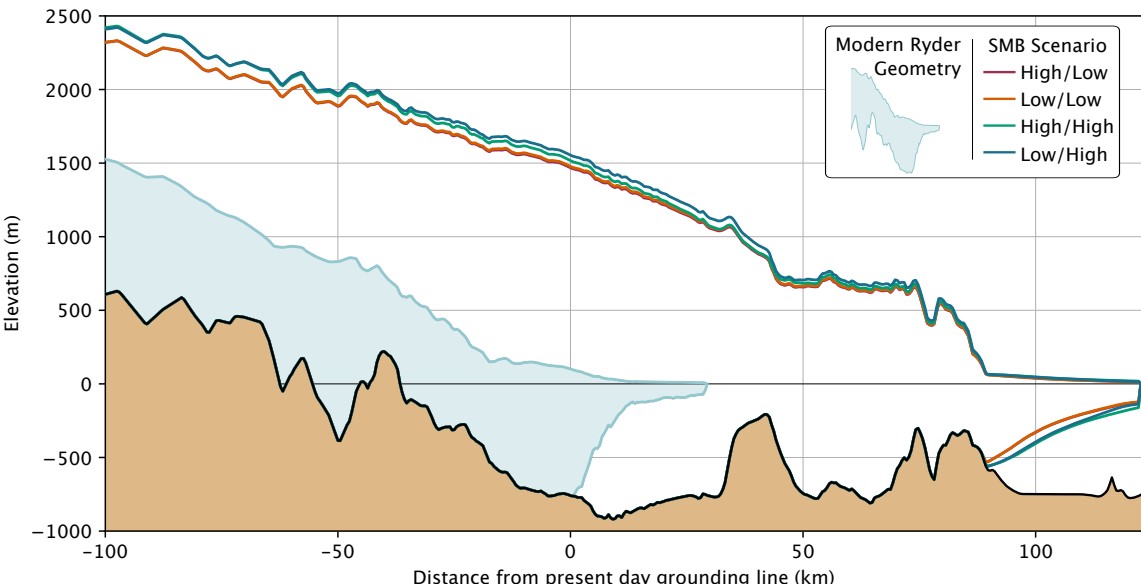

**Figure A1.** A comparison between the present day state of Ryder Glacier (light blue) and that of the glacier after running a 20,000 year spin-up forced with a YD climate for the differing SMB scenarios: High/Low (red), Low/Low (Orange), High/High (Green) and Low/High (Blue). (a) The velocity profile along the glacier's centre flowline. (b) A 2D cross section profile along the centre flowline. Bedrock data and the present day ice surface is from BedMachine v5 (Morlighem et al., 2022).





**Figure A2.** Evolution of Ryder Glacier in 2D cross section views along the centre flowline during the Holocene at specific time stamps. Dates of 10.3 ka **(a-d)**, 8.9 ka **(e-h)**, 3.9 ka **(i-l)** and 0 ka **(m-p)**, represent specific landmarks in the evolution of Ryder as discussed in O'Regan et al. (2021). Each row represents a different ocean forcing used in the transient simulation (Table 1), from top down this includes: *Reference*, *Calving_High*, *Calving_Low* and both extreme *High* and *Low* Scenarios. Each plot contains the 4 different SMB scenarios: High/Low (red), Low/Low (Orange), High/High (Green) and Low/High (Blue).



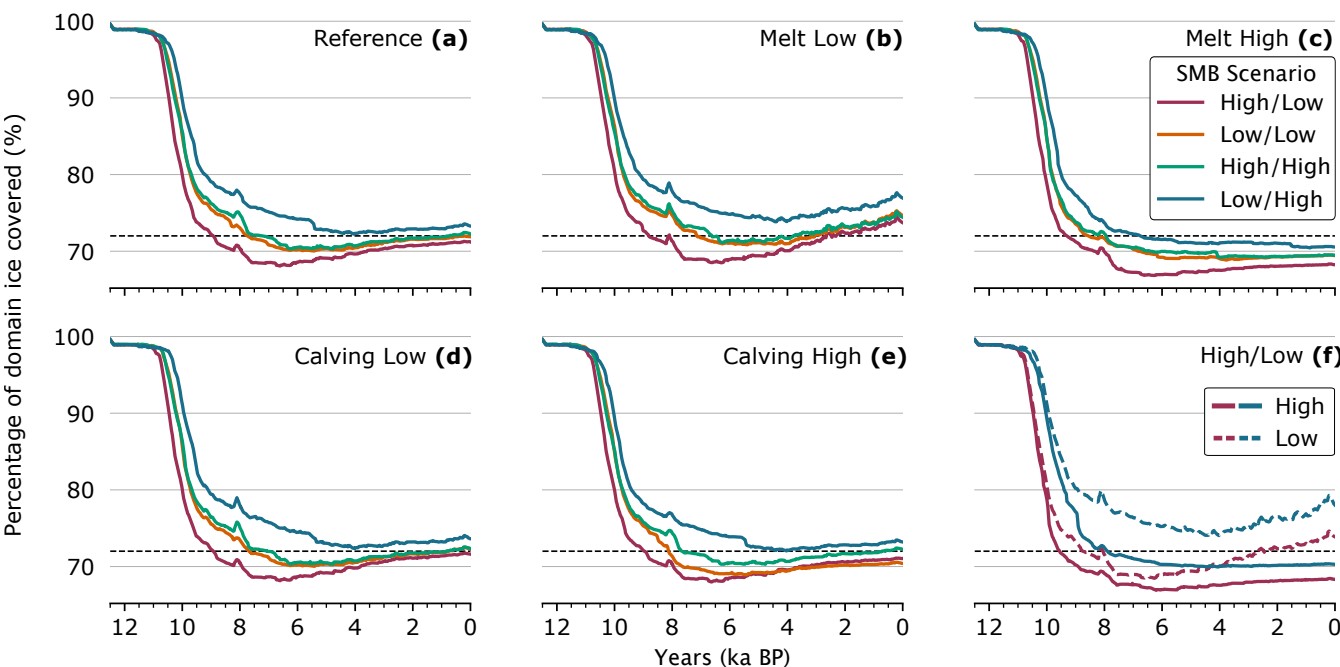

**Figure A3.** Evolution of ice cover in the model domain during the Holocene simulations. Line colour defines the SMB scenario for the transient runs: High/Low (red), Low/Low (Orange), High/High (Green) and Low/High (Blue). Each subplot represents a different ocean scenario: **(a)** *Reference*, **(b)** *Melt_Low*, **(c)** *Melt_High*, **(d)** *Calving_Low*, **(e)** *Calving_High* and **(f)** both the extreme *High* and *Low* ocean forcings. The dashed black line represent the contemporary percentage of the domain covered with ice, 72%.



*Author contributions.* The study was conceived by JB and MJ. JB ran all model simulations with technical advice from FAH, JC, HA and MM. JB completed data analysis on all simulations with the interpretation of results discussed amongst all authors. The manuscript was written by JB with significant input from all authors.

*Competing interests.* The authors declare no competing interests.

*Acknowledgements.* All simulation were conducted using resources provided by the National Academic Infrastructure for Supercomputing in Sweden (NAISS) at the National Supercomputer Centre at Linköping University, Sweden, partially funded by the Swedish Research Council through grant agreement no. 2022-06725. JB was funded by VR Grant no. 2021-04512 and FAH was funded by Formas Grant no. 2021-01590, both awarded to MJ. HÅ was supported by the project JOSTICE from the Research Council of Norway (grant no. 302458) and
ERC-2022-ADG grant agreement No 01096057 GLACMASS from the European Research Council.



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
