# Peer review of "Simulating the Holocene evolution of Ryder Glacier, North Greenland"

_EGUsphere, 2025_

## Referee Comment (RC1)

**Review of Barnett et al. (2025) 'Simulating the Holocene evolution of Ryder Glacier, North Greenland'**

**Summary**
This study presents the results of an ensemble of model runs of Ryder Glacier in northern Greenland across the entirety of the Holocene, aiming to explore the drivers of its early-Holocene retreat and late-Holocene re-advance as derived from geomorphological and sedimentary records. The paper shows that the initial retreat of the glacier at the start of the Holocene was almost certainly driven by changes in SMB – probably both warming temperatures and reduced precipitation – whereas later retreat was chiefly due to oceanic forcing. Crucially, the paper demonstrates that the late-Holocene re-advance of the glacier can only be explained by changes in both atmospheric and oceanic conditions, and that the geometry of Sherard Osborn Fjord exerts a major local control on the glacier's behaviour.

I think this is a good, well-structured paper that convincingly addresses its titular subject. The ensemble of simulations captures the observed evolution of the glacier and the authors provide a thorough discussion of their findings, including sensible explanations for why the model does or doesn't match well with different observations. The figures are also well thought-out and clear, which greatly helps the reader understand what is happening.

I do, however, have a couple of major concerns that I would like to see addressed before publication with regards to the modelling strategy. This may be as simple as adding a few sentences explaining some choices the authors made, but as it stands, I don't fully comprehend why the authors chose to use the Blatter-Pattyn approximation when this is a situation that cries out for a full-Stokes treatment, nor how they can be sure their reference value of their calving parameter is correct for this particular glacier. More details are below and I hope it is a straightforward case of adding a little bit more explanation, rather than something more complicated!

Page and line numbers refer to those in the clean version of the submitted manuscript.

**Major Comments**
- Choice of model: This study is very heavily focused on correctly modelling the grounding line and retreat of Ryder Glacier. Why then did the authors choose to use the Blatter-Pattyn approximation of Stokes, which is not valid at the grounding line (not in hydrostatic equilibrium), rather than a full-Stokes formulation? At the very least, there needs to be some acknowledgement in the paper that this might be a problem or limitation.
- Sigma_max values: The authors use a reference value of sigma_max, the key parameter in their calving law, based on studies at neighbouring Petermann Glacier. However, sigma_max should be calibrated to each domain, as the 'correct' value for one glacier may or may not be the same as for another one. Now, Petermann and Ryder are reasonably similar, so it might well be fine, but they're not the same. And Ryder during the HTM or the YD or similar is definitely very different to Ryder now. I think the authors need to show that 300 kPa works as a reference value by showing that it reproduces observed contemporary behaviour at Ryder at the very least, before being able to assume that it's a reasonable choice. Or some more elaborate justification needs to be added beyond 'it works at Petermann'.
- Language: The authors will notice that a lot of my minor comments are to do with slightly infelicitous phrasings, typos and poor word choices, to the extent that I'm pointing it out here as a problem (I will also say that I only noted down the ones that really bothered me – I might recommend a thorough re-read before submitting the corrected version to make sure there aren't any others). I admit that I'm pickier about this than some, but there are quite a few cases where I found it impeded my understanding of the point the authors were trying to make. Really, I just want to highlight that it makes the entire review process much smoother if the authors pick these up before submitting the paper (and means reviewers will be better able to engage with the substantive points of the paper if they're not having to spend time puzzling over what the paper is actually saying).

**Minor Comments**
- p.1, l.17: 'analogies', not 'analogous'

- p.2, l.31: 'the ice sheet's'
- p.2, l.36: 'affect'
- p.4, l.64: 'Innuitian'
- p.4, l.77: Remove the 's on the end of Glacier
- p.6, l.112: Blatter-Pattyn isn't valid at grounding lines, though, and this is a study largely focused on the grounding line of Ryder Glacier. This maybe seems a curious choice of approximation – why not use a full-Stokes setup?
- p.6, l.117: 'extent' not 'extend'
- p.6, l.119: OK, in Figure 2 it's 'Saint George Fjord', here it's 'Saint Georges Fjord' (and possibly should be 'Saint George's Fjord'). Which one is correct? Later in the paper, it's consistently 'St George Fjord', so I assume it's that one. Just make sure to be consistent.
- p.6, l.119: 'effect', not 'affect'
- Section 3.1: Maybe I missed it, but I can't see where the authors state what surface topography is being used to initialise the model? I assume it's also BedMachine (the caption to Figure 4 and Section 3.5 bear this out), but it should be stated clearly here too.
- p.7, l.154: Some confusion on dates here. If the simulations start in 12,500 BP, then today is year 0 and the simulations run for at most 12,500 years. If the simulations start in 12,500 BC, then running to AD 2000 makes a total of 14,500 years. Either way, I'm not quite sure how a total runtime of 12,550 years is achieved with the dates as written. Either put both dates in AD/BC (or CE/BCE, it's the same thing), or define when 'P' is in BP so that it's clear when the simulations actually start and how long they run for.
- p.9, l.198: Maybe put 'temperature and precipitation' in brackets to make the sentence a bit easier to read? Also, 'an 1850-2000 mean' here and on the next line.
- Figure 4: Might it be possible to extend the x-axis slightly farther (to 150, say)? As it stands, the YD glacier cross-section butts up right against the chart edge, which doesn't look great.
- p.10, l.213: Yes, fair enough, I'm sure it would be lower, but is there any justification for that reduced deepmelt parameter beyond a) it works and b) it's lower? It's the spin-up, it probably doesn't matter that much, but there maybe needs to be a bit more effort here to justify the value. I assume it's the lowest value of melt recorded in the observations and modelling, following Section 3.3, but it bears restating clearly here why the choice was made.
- p.11, l.233: I think 'where' is meant to be 'were', and the comma immediately before it should be deleted, or the sentence doesn't make much sense.
- p.11, l.230-237: Yes, but Petermann isn't Ryder, so a sigma_max value that works for Petermann may or may not be in any way correct for Ryder. Especially not Ryder at a different time in a fundamentally different set of climate conditions. Did the authors check that 300 kPa was a sensible reference value for Ryder by, for example, running some contemporary simulations to show the model reproduces observed behaviour at the glacier well with that value?
- p.12, l.253: 'comparison with' not 'comparison on'
- p.12, l.263: 'retreat', not 'retreating'
- p.17, l.310: 'sheer'
- p.17, l.321: 'increased'
- p.17, l.328: Should the second simulation be Low/High: *Calving Low*?
- p.17, l.330: I don't think you mean 'perceived' here. It's not a case of your perception being that the model has advanced; it's a model, either it's advanced or it hasn't. I might choose a different word.
- Figure 9 caption: 'set of' and 'used'
- p.18, l.371: 'For using' should probably just be 'For'?
- p.19, l.373: 'a final ice margin'
- Figure 10 caption: 'set of' and 'used'
- p.21, l.386: Just 'Ryder Glacier', not 'the Ryder Glacier'
- p.21, l.390: I'm not sure 'invoke' is the right word here. I think the authors mean 'cause', 'lead to', 'result in' or another synonym, of which 'invoke' is not one
- p.21, l.392-394: Delete the semicolon, replace it with a comma, and then replace 'that ranges' with 'ranging'
- p.21, l.395-401: The comparison to Cuzzone et al. (2022) is nice, but at the same time I could summarise this section as 'two different glaciers in very different settings exposed to very different environmental conditions behave differently', which isn't much of a surprise. I would suggest

removing it, or reducing it to a sentence along the lines of 'we expect these two glaciers to be different and they are', as I don't think it's really adding much to the discussion as written.

- p.21, l.407: Another interesting vocabulary choice: 'alluding to' is not the phrase required here; 'hinting at' may be more appropriate.
- p.21, l.412: 'in Baffin Bay and the Labrador Sea'
- p.21, l.417: 'ensuring the survival'
- p.22, l.427: 'invoked' was correctly used at l.414, but here it's not the right word again. It's not a synonym for 'cause' or similar, which is the sense intended here (if I'm parsing the sentence correctly).
- p.22, l.441: 'ostracod'
- p.23, l.452-453: I'm not sure I quite understand this as written. I think it's just a case of removing the comma after 'latter', but it may be the authors intended something else here.
- p.23, l.468-473: I may have missed some subtlety here, but why would anyone expect the inclusion or not of calving and ocean melt to have any effect on land-based ice in the first place? I would rephrase this to just talk about the similarity between this study and the Cuzzone paper with regards to the effect of including calving and ocean melt on the marine-terminating margins. Or the authors need to add some text explaining why either process would affect ice not touching the ocean, thus making the comparison worthwhile, which may be harder.
- p.23, l.477: 'inland', not 'in land', is I think what is meant?
- p.23, l.479-480: 'where retreat as calving cliff face produced exaggerated retreat' is a phrase I'm not able to draw much sense from, to the extent that I'm entirely sure what to suggest as an alternative phrasing. Please have a look and rephrase.
- p.23, l.482: 'aid', not 'aide' (aide is the noun form. Or French)
- p.23, l.484: Delete 'the' before 'Ryder's'
- p.24, l.486: Strictly speaking, 'protracted' does just mean 'lengthy', but it always carries a negative connotation (one can have 'a protracted meeting', but not 'a protracted party' unless one is really not enjoying oneself), which doesn't quite work here – a protracted timescale would be one that was unusually long compared to what was normal, whereas here the sense intended is just 'a long time' for something that is actually a long time. I'd replace it with 'lengthy' or 'extended'
- p.24, l.489: 'it's very likely'
- p.24, l.494: 'the ice tongue'
- p.24, l.500: 'to re-form'
- p.24, l.503: 'stress' – 'findings' is plural
- p.24, l.504: nope, that's not a semicolon – it should just be a comma
- p.24, l.518: 'implemented in'
- p.25, l.523: 'sit' – there are two things there
- p.25, l.527: 'with the latter able to weaken the floating'
- p.25, l.532: 'that is also shielded from warm AW by a bathymetric high'
- p.25, l.535-536: 'that will lead to a greater transfer of heat from the ocean to the glacier'
- p.25, l.536-537: 'set to play'
- p.25, l.539: 'relatively'
- p.25, l.548: 'that resembles that of the mid-Holocene'
- p.26, l.555: Delete either 'at' or 'with'
- p.26, l.557-558: 'The retreat...is'
- p.26, l.560-563: See my earlier comment. Finding that the terrestrial margin is insensitive to what's going on in the ocean is not really a significant finding. I would just focus this point on the terrestrial margin's position being narrowly linked to SMB
- p.26, l.564: 'the marine margin'
- Figure A1: What are (a) and (b) referring to in the caption? There's only one panel….

---

## Author Comment (AC1)

**Reviewer 1**

Summary

This study presents the results of an ensemble of model runs of Ryder Glacier in northern Greenland across the entirety of the Holocene, aiming to explore the drivers of its early-Holocene retreat and late-Holocene re- advance as derived from geomorphological and sedimentary records. The paper shows that the initial retreat of the glacier at the start of the Holocene was almost certainly driven by changes in SMB – probably both warming temperatures and reduced precipitation – whereas later retreat was chiefly due to oceanic forcing. Crucially, the paper demonstrates that the late-Holocene re-advance of the glacier can only be explained by changes in both atmospheric and oceanic conditions, and that the geometry of Sherard Osborn Fjord exerts a major local control on the glacier's behaviour.

I think this is a good, well-structured paper that convincingly addresses its titular subject. The ensemble of simulations captures the observed evolution of the glacier and the authors provide a thorough discussion of their findings, including sensible explanations for why the model does or doesn't match well with different observations. The figures are also well thought-out and clear, which greatly helps the reader understand what is happening.

I do, however, have a couple of major concerns that I would like to see addressed before publication with regards to the modelling strategy. This may be as simple as adding a few sentences explaining some choices the authors made, but as it stands, I don't fully comprehend why the authors chose to use the Blatter-Pattyn approximation when this is a situation that cries out for a full-Stokes treatment, nor how they can be sure their reference value of their calving parameter is correct for this particular glacier. More details are below and I hope it is a straightforward case of adding a little bit more explanation, rather than something more complicated!

**Thank you for the constructive review of the manuscript. We appreciate your comments and have responded to them in detail below in blue.**

**Major Comments:**

Choice of model: This study is very heavily focused on correctly modelling the grounding line and retreat of Ryder Glacier. Why then did the authors choose to use the Blatter-Pattyn approximation of Stokes, which is not valid at the grounding line (not in hydrostatic equilibrium), rather than a full- Stokes formulation? At the very least, there needs to be some acknowledgement in the paper that this might be a problem or limitation.

We agree with the reviewer that Ryder Glacier would ideally be deserving of the Full-Stokes (FS) treatment and appreciate the comment. However, as FS is of orders of magnitude more computationally expensive than the Blatter-Pattyn Higher-Order (HO) approximation, running FS over extended paleo time scales such as the Holocene is not possible today. We believe that HO offers the next best solution, as it maintains thermomechanical 3D ice dynamics, thus allowing for the changes in ice temperature to influence ice viscosity and retaining higher-order stress which is not possible in other approximations such as SSA or SIA. To address this point, we have added to our methods section to explain why we chose to

pursue a HO model set up, which, at spatial resolutions better than 1km, remains a novel approach to simulation ice dynamics over the Holocene in Greenland.

Re-written model set up:

*The finite-element thermomechanical ice-sheet model ISSM is employed to simulate the Holocene history of Ryder Glacier (Larour et al., 2012) together with a Higher-Order (HO) approximation (Blatter, 1995; Pattyn, 2003) of the Full Stokes (FS) equations. We use the HO approximation as it balances computational efficiency with thermomechanical 3D ice dynamics, including higher-order stresses and temperature-dependent viscosity, which are critical when simulating ice evolution over paleo-timescale. Although we recognise that FS offers greater accuracy in regions such as the grounding line, where the inclusion of bridging stresses, neglected in the HO approximation, are important, it remains computationally infeasible to use over extended timescales such as the Holocene. To better capture grounding migration, we run our HO model at resolutions of < 1km, following a model setup that is similar to Briner et al. (2020) and Cuzzone et al. (2022) which performed well when simulating both terrestrial and marine regions of the GrIS over the Holocene.*

Sigma_max values: The authors use a reference value of sigma_max, the key parameter in their calving law, based on studies at neighbouring Petermann Glacier. However, sigma_max should be calibrated to each domain, as the 'correct' value for one glacier may or may not be the same as for another one. Now, Petermann and Ryder are reasonably similar, so it might well be fine, but they're not the same. And Ryder during the HTM or the YD or similar is definitely very different to Ryder now. I think the authors need to show that 300 kPa works as a reference value by showing that it reproduces observed contemporary behaviour at Ryder at the very least, before being able to assume that it's a reasonable choice. Or some more elaborate justification needs to be added beyond 'it works at Petermann'.

Thank you for raising this uncertainty regarding calving laws and their associated tuning values. We agree that that the tuning of the sigma_max value for the Von Mises law is particular to each glacier in their respective current setting and therefore is likely to vary over extended timescales such as the Holocene. To address this, we have tested our reference sigma_max value of 300 kPa at the contemporary Ryder Glacier and found it to replicate the current terminus position well, this is demonstrated in the Figure below which will be added to the supplementary information of the manuscript. Furthermore, we have re-written the section about calving in our Methodology, where we explicitly state that the *Reference* tuning value is unlikely to remain valid during the entire Holocene, and therefore why we have run transient scenarios with different calving thresholds.

Re-written Methods:
*Our Reference calving threshold for floating ice is 300 kPa as this value was able to reproduce the recent evolution of Ryder Glacier's floating tongue (Fig. A1) and aligns with the threshold used in a study of the neighbouring Petermann glacier by Åkesson et al. (2022). Because calving laws are often calibrated for glaciers in specific states, influenced by factors such as fjord geometry, bedrock topography, and ice rheology (Amaral et al., 2020; Wilner et al., 2023), we acknowledge that our chosen Reference calving threshold is unlikely to remain valid throughout the entire Holocene. To evaluate the influence of the calving threshold on ice dynamics, we conduct transient simulations with varied threshold values. For our High_Calving scenario, we lower the threshold to be 200 kPa …*

Additional calving threshold test – Figure and text to be included in Supplementary Material:

[Figure]

*Figure A1. A comparison between the recent observed evolution of Ryder's calving front with a simulated front using the Von Mises Calving Law. The 2008 position of Ryder's terminus (black) is sourced from Bedmachine v3 (Morlighem et al., 2017) and is taken as the initial ice front position for our comparison simulation. The 2025 position of Ryder's terminus (red) is taken from a Sentinel 2 satellite image dated 19-05-2025. The simulated 2025 position of Ryder (dashed orange) uses the Von Mises calving law with a threshold of 300 kPa. Surface topography and ocean bathymetry are from BedMachine v5 (Morlighem et al., 2022).*

*Here we test whether the Von Mises calving law with σmax value of 300 kPa (Eq. 4), found to be a suitable tuning value for the neighbouring Petermann Glacier (Åkesson et al., 2022), reproduces the contemporary evolution of Ryder Glacier's terminus. To do this, we run our ice sheet model from the year 2008 forward to 2025 and compare the position of the ice front to observations. The initial ice front position and ice geometry are taken from Bedmachine v3 which has a nominal date of 2008 (Morlighem et al., 2017). Figure A1 shows the position of the observed front in 2025 against our simulated front using the Von Mises law with a σmax value of 300 kPa. The calving law reproduces the 2025 position well capturing the overall trend of glacier advance during the 17 years. While the centre of the ice tongue terminus is slightly too advanced, its margins, and their connection to the fjord walls which offer buttressing, terminate in a similar position to that observed in 2025.*

Language: The authors will notice that a lot of my minor comments are to do with slightly infelicitous phrasings, typos and poor word choices, to the extent that I'm pointing it out here as a problem (I will also say that I only noted down the ones that really bothered me – I might recommend a thorough re-read before submitting the corrected version to make sure there aren't any others). I admit that I'm pickier about this than some, but there are quite a few cases where I found it impeded my understanding of the point the authors were trying to make. Really, I just want to highlight that it makes the entire review process much smoother if the authors pick these up before submitting the paper (and means reviewers will be better able to engage with the substantive points of the paper if they're not having to spend time puzzling over what the paper is actually saying).

We thank the reviewer one for their comments regarding grammar/language and have incorporated all their suggested changes which are outlined below in the minor comments sections. After a further proofread we believe that the manuscript now reads much better.

**Minor Comments:**

p.1, l.17: 'analogies', not 'analogous'
Corrected
p.2, l.31: 'the ice sheet's'
Corrected
p.2, l.36: 'affect'
Corrected
p.4, l.64: 'Innuitian'
Corrected
p.4, l.77: Remove the 's on the end of Glacier
Corrected and re-wrote the sentence.
p.6, l.112: Blatter-Pattyn isn't valid at grounding lines, though, and this is a study largely focused on the grounding line of Ryder Glacier. This maybe seems a curious choice of approximation – why not use a full-Stokes setup?
We have addressed this in our response to the major comment above, but reiterate that FS over the Holocene at such high resolutions is not yet computationally possible.
p.6, l.117: 'extent' not 'extend'
Amended and sentence reworked
p.6, l.119: OK, in Figure 2 it's 'Saint George Fjord', here it's 'Saint Georges Fjord' (and possibly should be 'Saint George's Fjord'). Which one is correct? Later in the paper, it's consistently 'St George Fjord', so I assume it's that one. Just make sure to be consistent.
Thanks for spotting this. It is now St. George Fjord everywhere.
p.6, l.119: 'effect', not 'affect'
Corrected
Section 3.1: Maybe I missed it, but I can't see where the authors state what surface topography is being used to initialise the model? I assume it's also BedMachine (the caption to Figure 4 and Section 3.5 bear this out), but it should be stated clearly here too.
We have added the following line to Section 3.1 to address this:
*The initial ice surface for our model is also taken from Bedmachine v5, before it evolves in our Younger Dryas spin-up (Section 3.5).*
p.7, l.154: Some confusion on dates here. If the simulations start in 12,500 BP, then today is year 0 and the simulations run for at most 12,500 years. If the simulations start in 12,500 BC, then running to AD 2000 makes a total of 14,500 years. Either way, I'm not quite sure how a total runtime of 12,550 years is achieved with the dates as written. Either put both dates in

AD/BC (or CE/BCE, it's the same thing), or define when 'P' is in BP so that it's clear when the simulations actually start and how long they run for.

Thanks for raising this, we understand why this brought up some confusion. We have defined P as being 1950 as follows:

*All model simulations are run for 12,550 years from 12,500 BP, where P is defined as 1950 CE, to 2000 CE…*

p.9, l.198: Maybe put 'temperature and precipitation' in brackets to make the sentence a bit easier to read? Also, 'an 1850-2000 mean' here and on the next line.

This has been amended.

Figure 4: Might it be possible to extend the x-axis slightly farther (to 150, say)? As it stands, the YD glacier cross-section butts up right against the chart edge, which doesn't look great.

We have extended the x-axis to 150 km and agree it does look better. Thank you.

p.10, l.213: Yes, fair enough, I'm sure it would be lower, but is there any justification for that reduced deepmelt parameter beyond a) it works and b) it's lower? It's the spin up, it probably doesn't matter that much, but there maybe needs to be a bit more effort here to justify the value. I assume it's the lowest value of melt recorded in the observations and modelling, following Section 3.3, but it bears restating clearly here why the choice was made.

Thanks for highlighting this. We have re-written our Section 3.5: Younger Dryas Spin Up to better articulate why the 10m/yr deep melt rate is an appropriate choice, highlighting how it position of the grounding line and ice shelf fit with geological record

*… We do not allow calving during the spin-up. Ocean melt is applied after the first 2,000 years, in order to allow for grounding line advance, after which the deepmelt rate is set to 10 m/yr. While an arbitrary choice, we argue that ocean melt at the YD would be substantially lower than contemporary rates at Ryder Glacier (Wilson et al., 2017), due to both colder ocean temperatures and a reduction in subglacial discharge in a climate with little surface melt (Fig. 3a). The combination of ocean melt and atmospheric forcings results in a spun-up Ryder Glacier that aligns with the geological record at the YD, with the grounding line resting atop the outer sill where the oldest sediments recovered from Sherard Osborn Fjord are dated to 12 ± 0.5 ka BP (Fig. 4; O'Regan et al., 2021). The stable ice shelf emanating from the fjord also matches the hypothesised Lincoln Sea Ice Stream and ice shelf conditions across Northern Greenland (Funder and Larsen, 1982; Dawes, 1986; Larsen et al., 2010)…*

p.11, l.233: I think 'where' is meant to be 'were', and the comma immediately before it should be deleted, or the sentence doesn't make much sense.

This has been amended and the sentence re-written

*For our High_Calving scenario, we lower the threshold to 200 kPa to implicitly account for an extended calving period during the HTM when sea ice conditions were more seasonal (Detlef et al., 2023).*

p.11, l.230-237: Yes, but Petermann isn't Ryder, so a sigma_max value that works for Petermann may or may not be in any way correct for Ryder. Especially not Ryder at a different time in a fundamentally different set of climate conditions. Did the authors check that 300 kPa was a sensible reference value for Ryder by, for example, running some contemporary simulations to show the model reproduces observed behaviour at the glacier well with that value?

Thank you again for raising this. Please see our response to the same point in the major comments section above which we hope is satisfactory.

p.12, l.253: 'comparison with' not 'comparison on'

Fixed.

p.12, l.263: 'retreat', not 'retreating'

Fixed this other grammar mistakes in the same paragraph.

p.17, l.310: 'sheer'

Corrected.

p.17, l.321: 'increased'

Corrected.

p.17, l.328: Should the second simulation be Low/High: Calving Low?

The second simulation is correct being Low/High:*Low*.

p.17, l.330: I don't think you mean 'perceived' here. It's not a case of your perception being that the model has advanced; it's a model, either it's advanced or it hasn't. I might choose a different word.

True, we have adjusted the sentence:

*In runs where the grounding line does not readvance …*

Figure 9 caption: 'set of' and 'used'

We have correct this and re-structured the sentence in the figure caption. The same has been done for Figure 10.

p.18, l.371: 'For using' should probably just be 'For'?

Yes, corrected.

p.19, l.373: 'a final ice margin'

This sentence has been re-written.

Figure 10 caption: 'set of' and 'used'

Fixed just like Figure 9's caption.

p.21, l.386: Just 'Ryder Glacier', not 'the Ryder Glacier'

Fixed.

p.21, l.390: I'm not sure 'invoke' is the right word here. I think the authors mean 'cause', 'lead to', 'result in' or another synonym, of which 'invoke' is not one

Invoke has been changed to lead to.

p.21, l.392-394: Delete the semicolon, replace it with a comma, and then replace 'that ranges' with 'ranging'

Thank you, this reads better.

p.21, l.395-401: The comparison to Cuzzone et al. (2022) is nice, but at the same time I could summarise this section as 'two different glaciers in very different settings exposed to very different environmental conditions behave differently', which isn't much of a surprise. I would suggest removing it, or reducing it to a sentence along the lines of 'we expect these two glaciers to be different and they are', as I don't think it's really adding much to the discussion as written.

This makes sense, we have tidied up the paragraph by reducing the comparison with Cuzzone et al. (2022) to a sentence at the end.

*… these results contrast Holocene modelling studies of Southwestern Greenland, where variations in temperature played a greater role in determining retreat rate (Briner et al., 2020; Cuzzone et al., 2022).*

p.21, l.407: Another interesting vocabulary choice: 'alluding to' is not the phrase required here; 'hinting at' may be more appropriate.

'Hinting at' is better suited here. We have corrected.

p.21, l.412: 'in Baffin Bay and the Labrador Sea'

Fixed.

p.21, l.417: 'ensuring the survival'

Added the word 'ensuring'.

p.22, l.427: 'invoked' was correctly used at l.414, but here it's not the right word again. It's not a synonym for 'cause' or similar, which is the sense intended here (if I'm parsing the sentence correctly).

'Invoked' has been changed to 'leads to'

p.22, l.441: 'ostracod'

Spelling is fixed.

p.23, l.452-453: I'm not sure I quite understand this as written. I think it's just a case of removing the comma after 'latter', but it may be the authors intended something else here.

We have removed this sentence entirely and believe the paragraph now works better.

p.23, l.468-473: I may have missed some subtlety here, but why would anyone expect the inclusion or not of calving and ocean melt to have any effect on land-based ice in the first place? I would rephrase this to just talk about the similarity between this study and the Cuzzone paper with regards to the effect of including calving and ocean melt on the marine-terminating margins. Or the authors need to add some text explaining why either process would affect ice not touching the ocean, thus making the comparison worthwhile, which may be harder.

This is a good point, and we understand the why the reviewer has made this comment. Our intention was to highlight how Cuzzone (2022) found the inclusion of calving which resulted in an increase in inland thinning did not cause the terrestrial margin retreat any faster. So, there was no **in-direct** impact on the terrestrial margin by including calving that thinned the interior. When re-reading the manuscript, we found that removing the discussion on terrestrial ice made the paragraph read better. Therefore, following the reviewers suggestion, we now just talk about the similarities between our paper and Cuzzone (2022):

*Cuzzone et al. (2022) note how the inclusion of calving allowed the marine margins to persist longer in the fjords, relative to adjacent terrestrial ice, due to a greater transport of mass from the interior, which in turn led to thinning in the upper regions of the domain. We find such consistencies across our simulations during the early Holocene, where, despite rapid retreat of the terrestrial ice, Ryder Glacier maintains an extended position in the fjord (Fig. 9).*

p.23, l.477: 'inland', not 'in land', is I think what is meant?

Yes, typo fixed.

p.23, l.479-480: 'where retreat as calving cliff face produced exaggerated retreat' is a phrase I'm not able to draw much sense from, to the extent that I'm entirely sure what to suggest as an alternative phrasing. Please have a look and rephrase.

We agree that this statement was not as clear as it should have been and thank the reviewer for noticing. We have re-written the sentence to better describe the shift in calving once the floating tongue is lost:

*When we simulate a collapse of the ice tongue, either during retreat from the inner sill (Fig. 7) or at Ryder's Holocene minimum (Fig. 8), the loss of buttressing and transition to calving from an unconstrained grounded front leads to an increase in ice discharge and inland thinning.*

p.23, l.482: 'aid', not 'aide' (aide is the noun form. Or French)

Corrected
p.23, l.484: Delete 'the' before 'Ryder's'
Removed 'the'.
p.24, l.486: Strictly speaking, 'protracted' does just mean 'lengthy', but it always carries a negative connotation (one can have 'a protracted meeting', but not 'a protracted party' unless one is really not
enjoying oneself), which doesn't quite work here – a protracted timescale would be one that was unusually long compared to what was normal, whereas here the sens intended is just 'a long time' for something that is actually a long time. I'd replace it with 'lengthy' or 'extended'
We have changed 'protracted' to 'extended'.
p.24, l.489: 'it's very likely'
Fixed.
p.24, l.494: 'the ice tongue'
Added 'the'.
p.24, l.500: 'to re-form'
Added 'to'.
p.24, l.503: 'stress' – 'findings' is plural
Fixed.
p.24, l.504: nope, that's not a semicolon – it should just be a comma
Changed to a comma.
p.24, l.518: 'implemented in'
Corrected.
p.25, l.523: 'sit' – there are two things there
Fixed.
p.25, l.527: 'with the latter able to weaken the floating'
Spellings and grammar fixed.
p.25, l.532: 'that is also shielded from warm AW by a bathymetric high'
Corrected.
p.25, l.535-536: 'that will lead to a greater transfer of heat from the ocean to the glacier'
Corrected this sentence
 p.25, l.536-537: 'set to play;
Fixed.
p.25, l.539: 'relatively'
Corrected.
p.25, l.548: 'that resembles that of the mid-Holocene'
Corrected the wording here.
p.26, l.555: Delete either 'at' or 'with'
Deleted with.
p.26, l.557-558: 'The retreat...is'
Fixed.
p.26, l.560-563: See my earlier comment. Finding that the terrestrial margin is insensitive to what's going on in the ocean is not really a significant finding. I would just  focus this point on the terrestrial margin's position being narrowly linked to SMB
Thank you again for raising this. We have changed this sentence in our conclusion to focus on the terrestrial margins link to precipitation:

*Precipitation rates played a key role in modulating both the pace and magnitude of retreat, likely reflecting the arid nature of Northern Greenland. We find that the climate scenario*

*where mid-Holocene precipitation rates exceed that of contemporary levels produce terrestrial ice retreat that best matches the geologic record.*

p.26, l.564: 'the marine margin'
Fixed.
Figure A1: What are (a) and (b) referring to in the caption? There's only one panel….
The figure caption has been fixed to describe the single panel.

---

## Author Comment (AC2)

Reviewer 2:

**Manuscript Summary**

This manuscript presents the results of ISSM simulations of Ryder Glacier, North Greenland, through the Holocene, constrained by preexisting terrestrial reconstructions and marine sediment records. They aim to better understand the drivers of Holocene ice sheet retreat and advance. Their results show that retreat following the Younger Dryas was SMB driven, with ice-ocean interactions becoming more important during the mid-Holocene. They also demonstrate that the Neoglacial advance of Ryder Glacier requires ice tongue regrowth, which required both cooling ocean temperatures and air temperatures.

The manuscript is well-written and structured, effectively addressing the paper's objectives. It is very well illustrated, which helps me (and any reader) to understand what is being presented. The interpretations are clear and well supported by the results presented.

I don't have many major criticisms or concerns with the paper, but I think some aspects of the writing should be improved. There are numerous typos and incorrect word choices which limit the understanding of the content at times. I've included some below, but there are more.

**Thank you for taking the time to read and comment on our manuscript. We have responded to each of the points individually below in blue.**

Minor comments:

The decision behind the choice of model is not very clearly explained. Given the dependence of the paper on this, and as a non-modelling expert, I feel it would be beneficial to explain the choices for this glaciological and topographic situation.

We thank the reviewer for highlighting this section of the manuscript and note that Reviewer #1 had a similar point regarding model set up. To address this, we have re-written our initial paragraph in our methods section that describes why we chose to use the Blatter-Pattyn Higer-Order (HO) approximation for our study and not the Full-Stokes (FS) equations.

Re-written model set up:

*The finite-element thermomechanical ice-sheet model ISSM is employed to simulate the Holocene history of Ryder Glacier (Larour et al., 2012) together with a Higher-Order (HO) approximation (Blatter, 1995; Pattyn, 2003) of the Full Stokes (FS) equations. We use the HO approximation as it balances computational efficiency with thermomechanical 3D ice dynamics, including higher-order stresses and temperature-dependent viscosity, which are critical when simulating ice evolution over paleo-timescale. Although we recognise that FS offers greater accuracy in regions such as the grounding line, where the inclusion of bridging stresses, neglected in the HO approximation, are important, it remains computationally infeasible to use over extended timescales such as the Holocene. To better capture grounding migration, we run our HO model at resolutions of < 1km, following a model setup that is similar to Briner et al. (2020) and Cuzzone et al. (2022) which performed well when simulating both terrestrial and marine regions of the GrIS over the Holocene.*

You discuss the Steensby Stade as the last part of the Holocene glacial history of North Greenland, and also discuss the late Holocene advance, which reached its maximum at the LIA. Are these the same advance – i.e. both culminated in the LIA? I think some clarity should be given about this – including the LIA limit. In Fig 2a you label the two previous Stades, and in Fig. 2b you have the LIA (offshore) limit labelled. Is this LIA actually limit actually recorded, I had a quick look at the Koch (1928) reference and couldn't see how this limit was recorded. Is there an onshore expression of the LIA too?

Thank you for raising this point. We are happy to clear up that the Steensby Stade, as described by Kelly and Bennike (1992), represents the re-advance of the GrIS from the mid-Holocene minimum to its recent maximum at the LIA. Regarding Koch (1928), there is a map show Ryder's extended ice tongue, its not possible to interpret the terrestrial extent of the ice clearly and is therefore not show in Figure 2. We have added text to our introduction and description of Ryder's Holocene behaviour to better describe the Steensby Stade and LIA.

Text added to the introduction:

*Finally, the Steensby Stade characterises the neo-glacial re-advance of the GrIS from its Holocene minimum towards its most recent maximum in the Little Ice Age (LIA).*

Ryder's re-advance in the late-Holocene to LIA:

*The following lithologic facies after 3.9 ± 0.4 ka BP (LU2 and LU1), inline with the Steensby Stade described by Kelly and Bennike (1992), show a transition from open-water bioturbated sediments to laminated facies that indicate the regrowth and advance of Ryder's ice tongue towards the outer sill. Lauge Koch mapped Ryder's LIA extent in 1917 during the Second Thule Expedition, showing the ice tongue had extended toward the mouth of Sherard Osborn Fjord and the outer sill (Fig. 2b).*

Younger Dryas spin up: you say that you are assuming that the Ryder Glacier was stable during the Younger Dryas. Is there any evidence to back this up? The YD in Greenland is very enigmatic, and there is often little evidence for a stillstand, moraine formation, or dramatic slowdown during it (unless I am mistaken). So, I'd slightly caution the statement that the glacier was stable at this point in time.

We appreciated the reviewer noting this it is a valid point to highlight how the YD was enigmatic and likely not consistent across GrIS, where large parts of the ice sheet had already retreated a fair distance from the shelf and in some cases into local fjords. Nevertheless, we would like to stand by our assumption that Ryder Glacier in particular appears stable during the YD. This stems from the oldest sediments in Sherard Osborn Fjord being dated to 12 ± 0.5 ka BP on the outer sill, which indicates Ryder was likely stable on the bathymetric high during the YD. It is also documented that retreat in the northern sector of GrIS was comparatively limited to other regions prior to the YD. We have included these points and

extra references in our Spin Up section which we think helps ground out assumption that a YD spin up is acceptable and thank the reviewer for the comment.

New Spin Up Section:

*We perform a Younger Dryas (YD) model spin-up to obtain an initial ice-sheet state that is both geologically consistent and internally coherent. We assume that Ryder Glacier remained stable during the YD, consistent with the earliest recovered sediments from Sherard Osborn Fjord (12 ± 0.5 ka BP), which indicate that the grounding line had not yet begun to retreat and was likely positioned on the outer sill (O'Regan et al., 2021). Although other sectors of the Greenland Ice Sheet (GrIS) began retreating from their Last Glacial Maximum (LGM) extents prior to the YD, retreat in the northern sector was comparatively limited (Leger et al., 2023). In this region, the grounding lines of major outlet glaciers remained stable at or near the fjord mouths, typically grounded on bathymetric highs, until the onset of early Holocene warming (Kelly and Bennike, 1992; England, 1999; Jennings et al., 2011; Jakobsson et al., 2018; Jennings et al., 2019; O'Regan et al., 2021).*

*We begin our spin-up simulations using an initial modern-day ice surface from BedMachine v5 (Morlighem et al., 2022) and force the model with a constant climate from 12,500 BP. We conduct a spin-up for the four different SMB scenarios…*

L154: 12,500 BP until 2000 CE is 14,500 years, not 12,500.

Thanks for raising this, we understand why this brought up some confusion. We have defined P as being 1950, and added the following:

*All model simulations are run for 12,550 years from 12,500 BP, where P is defined as 1950 CE, to 2000 CE…*

P230: Is using the calving threshold for Petermann glacier here valid? Especially as the Petermann simulation was for present day conditions.

We are grateful that you highlighted this, and we recognise that the calibration of calving laws represents one of the largest current uncertainties in ice sheet modelling. To address this, we have amended in manuscript in two ways. Firstly, we tested our reference sigma_max threshold at the contemporary Ryder Glacier and found it was able to capture the recent change in terminus position well. This is show in the figure below which will be added to the Supplementary Information. Second, we have re-written the section in our Methodology where we discuss the calving thresholds. Here we discuss why a calving law calibrated to present day is likely not valid during the entire Holocene which is why we have used three different thresholds in our transient experiments.

Re-written Methods:
*Our Reference calving threshold for floating ice is 300 kPa, this value was able to reproduce the recent evolution of Ryder Glacier's floating tongue (Fig. A1) and aligns with the threshold used in a study of the neighbouring Petermann glacier by Åkesson et al. (2022).*

*Because calving laws are often calibrated for glaciers in specific states, influenced by factors such as fjord geometry, bedrock topography, and ice rheology (Amaral et al., 2020; Wilner et al., 2023), we acknowledge that our chosen Reference calving threshold is unlikely to remain valid throughout the entire Holocene. To evaluate the influence of the calving threshold on ice dynamics, we conduct transient simulations with varied threshold values. For our High_Calving scenario, we lower the threshold to be 200 kPa …*

Additional calving threshold test:

[Figure]

*Figure A1. A comparison between the recent observed evolution of Ryder's calving front with a simulated front using the Von Mises Calving Law. The 2008 position of Ryder's terminus (black) is sourced from Bedmachine v3 (Morlighem et al., 2017) and is taken as the initial ice front position for our comparison simulation. The 2025 position of Ryder's terminus (red) is taken from a Sentinel 2 satellite image dated 19-05-2025. The simulated 2025 position of Ryder (dashed orange) uses the Von Mises calving law with a threshold of 300 kPa. Surface topography and ocean bathymetry are from BedMachine v5 (Morlighem et al., 2022).*

Here we test whether the Von Mises calving law with σmax value of 300 kPa (Eq. 4), found to be a suitable tuning value for the neighbouring Petermann Glacier (Åkesson et al., 2022), reproduces the contemporary evolution of Ryder Glacier's terminus. To do this, we run our ice sheet model from the year 2008 forward to 2025 and compare the position of the ice front to observations. The initial ice front position and ice geometry are taken from Bedmachine v3 which has a nominal date of 2008 (Morlighem et al., 2017). Figure A1 shows the position of the observed front in 2025 against our simulated front using the Von Mises law with a σmax value of 300 kPa. The calving law reproduces the 2025 position well capturing the overall trend of glacier advance during the 17 years. While the centre of the ice tonuge terminus is slightly too advanced, the margins of the ice shelf, and their connection to the fjord walls which offer buttressing, terminate in a similar position to that observed in 2025.

Section 5.3 – the recent work on Nioghalvfjerdsbræ (79N) is partially included here (Smith et al. 2023), but further work has been completed here, in a similar setting (floating ice tongue collapse). Some further discussion of differences or similarities in forcings could be pertinent, as there is both offshore and onshore data.

Thank you for highlighting this, you are right to mention the new and interesting work being undertaken at 79N regarding its deglaciation from the continental shelf at the LGM to present day. We have added the modelling study of the Northeast Greenland Ice Stream by Tabone et al (2024) to our discussion on ocean melt rates that stress the importance of rising ocean temperatures in driving retreat from stable bathymetric positions.

L468: This may just be due to strange phrasing, but calving wouldn't have an impact on the terrestrial ice margin, unless you mean indirect impact in comparison to adjacent marine areas?

We appreciate you highlight this confusion, and note this was a similar comment to that raised by Reviewer #1. Our intention was to highlight how Cuzzone (2022) found the inclusion of calving and the resulting increase in inland thinning did not cause in the terrestrial margin retreat any faster. So, there was no **in-direct** impact on the terrestrial margin by including calving that thinned the interior. When re-reading the manuscript, we found that removing the discussion on terrestrial ice made the paragraph read better. Therefore, we now just talk about the similarities between our paper and Cuzzone (2022):

*Cuzzone et al. (2022) note how the inclusion of calving allowed the marine margins to persist longer in the fjords, relative to adjacent terrestrial ice, due to a greater transport of mass from the interior, which in turn led to thinning in the upper regions of the domain. We find such consistencies across our simulations during the early Holocene, where, despite rapid retreat of the terrestrial ice, Ryder Glacier maintains an extended position in the fjord (Fig. 9).*